# PROVABLE WEAK-TO-STRONG GENERALIZATION VIA OVERSPECIFIED STUDENTS AND UNDERSPECIFIED TEACHERS

## ABSTRACT

Weak-to-strong generalization, as introduced in Burns et al. (2023), describes the phenomenon that a strong student (e.g., GPT-4) trained solely on labels generated by a weaker teacher (e.g., GPT-2) can outperform the teacher's performance.

In this work, we study the underlying mechanism behind weak-to-strong generalization in a controlled setting based on random feature models, specifically two-layer neural networks with random features and trainable linear output layers. We consider a regime where the teacher is *underspecified* and cannot recover the ground-truth function, while the student is *overspecified* and capable of achieving exact recovery. Our analysis reveals that the teacher's limited capacity leads to unavoidable errors in the subspace spanned by low-variance directions of the data covariance matrix, when the groundtruth function contains significant signal in these directions. In contrast, the student, when trained by gradient flow with fixed random features, exhibits slower convergence in these low-variance directions and hence can explicitly reduce the error of the weak teacher through early stopping. Therefore, the student can obtain improved generalization performance compared with the teacher and enjoy weak-to-strong generalization. Finally, we provide a theoretical characterization of the spectral conditions of the data covariance matrix, under which weak-to-strong generalization provably occurs.

## 1 INTRODUCTION

Large-scale foundation models have strong capabilities across various domains, such as understanding vision and language tasks (Brown et al., 2020; Dosovitskiy et al., 2020; Radford et al., 2021; Chowdhery et al., 2023; OpenAI et al., 2023; Liu et al., 2023). Large models can sometimes exhibit complex behavior which humans cannot fully understand (Burns et al., 2023). Therefore, it becomes increasingly important to steer these large models to align with human preferences, despite the fact that humans may not be able to provide perfect supervision to these large models. This motivates the important goal of *superalignment* (Burns et al., 2023), which requires human with weak ability to supervise strong models such as large language models (LLMs).

Burns et al. (2023) showed that the goal of superalignment can be possibly achieved via *weak-to-strong generalization*, where a strong student (e.g., GPT-4) trained solely on labels generated by a weak teacher (e.g., GPT-2) can outperform the teacher's performance. Subsequent works have extended this idea by empirically investigating weak-to-strong generalization across a variety of architectures and training setups (Ji et al., 2024; Guo et al., 2024; Liu & Alahi, 2024; Yang et al., 2024b; Tao & Li, 2024; Zhou et al., 2025). Despite universal empirical observations of weak-to-strong generalization in the literature, the underlying mechanism of these observations remains elusive. Our paper is motivated by several key empirical observations from (Burns et al., 2023):

- (**E1**) Figure 2 of Burns et al. (2023) highlights a substantial performance gap between the weak teacher and the strong student when both are trained on ground-truth labels. In addition, the weak teacher performs uniformly poorly on various tasks.

- (**E2**) Figure 3 of Burns et al. (2023) reveals that weak-to-strong generalization in natural language processing (NLP) fine-tuning tasks improves consistently with increasing sizes of both student and teacher models;

- (**E3**) Figure 13 of Burns et al. (2023) indicates that a smaller student-to-teacher capacity ratio leads to better generalization, which can be achieved through early stopping during student training.

These observations suggest a nuanced interplay between model capacity, generalization, and training dynamics. To develop a theoretical insight into these interactions, we investigate weak-to-strong generalization in a controlled setting based on random feature models, specifically two-layer neural networks with random features and trainable linear output layers. We consider a setting where the ground-truth function is a $d$-dimensional linear model, the teacher is underspecified with $M_{\mathsf{TE}} < d$ hidden neurons and thus cannot recover the target function, while the student is overspecified with $M_{\mathsf{ST}} > d$ neurons and can achieve exact recovery. This formulation captures the capacity mismatch underlying observation (**E1**), which explains the performance gap between weak and strong models and the universally poor performance of the teacher model. Moreover, motivated by observations (**E2**) and (**E3**), our analysis investigates how the sizes of the teacher and student networks, as well as the dynamics of gradient-based training, govern the emergence of weak-to-strong generalization.

It is worth noting that a recent important work (Medvedev et al., 2025) also investigates weak-to-strong generalization using random feature models. However, their setting fundamentally differs from ours: both their teacher and student models are overspecified compared with the ground-truth function, meaning that with sufficient training on ground-truth labels, both models can in principle achieve zero loss. This idealized scenario does not align with practical observations—for example, Figure 2 in Burns et al. (2023) shows that even when trained on ground-truth labels, the weak teacher always achieves low accuracy on various tasks (e.g., 60% on the representative NLP task, 30% on the chess puzzle task). In contrast, our framework explicitly models an underspecified teacher and an overspecified student. This setup naturally explains the persistent performance gap and the weak teacher's poor performance even when trained with ground-truth supervision, as observed in practice (Burns et al., 2023). This asymmetry leads to a qualitatively different generalization regime and necessitates a fundamentally new theoretical analysis. Importantly, our setting better reflects real-world scenarios where stronger models are trained using supervision from weaker ones, enhancing the practical relevance of our results. The main contributions are summarized as follows.

- We derive explicit closed-form expressions for the prediction errors of both the teacher and the student networks within the random feature model framework. Our analysis reveals that the teacher exhibits large prediction errors along the low-variance directions of the data covariance matrix due to limited model capacity, when the groundtruth function contains significant signals in these directions.

- We show that the student, when trained by gradient flow with fixed random features, converges more slowly along low-variance directions of the data covariance matrix. This implicit bias enables the student to reduce the error of the weak teacher through early stopping, while still accurately capturing the dominant signal components. The key novelty in our analysis is to leverage the eigenlearning framework (Simon et al., 2023) to characterize the learning dynamics underlying this weak-to-strong generalization.

- We provide a theoretical characterization of the spectral conditions of the data covariance matrix, under which weak-to-strong generalization provably occurs. We also provide a lower bound for a general eigenvalue distribution for the data covariance matrix. Our results identify a key factor: a larger teacher model size leads to stronger weak-to-strong generalization. This extends the findings of (Medvedev et al., 2025) to our setting.

- We provide experimental results to validate our theoretical predictions and explore settings outside the scope of our theory, suggesting possible avenues for future research. These experiments demonstrate the occurrence of weak-to-strong generalization with various data distributions, model architectures, and model sizes.

## 2 RELATED WORK

**Theory of Weak to Strong Generalization.** Lang et al. (2024) analyzed weak-to-strong generalization based on assumption that the strong student model cannot learn the mistakes of the weak teacher due to an expansion condition on the population data distribution. Shin et al. (2024) built off the framework in Lang et al. (2024) and considered the overlap density of easy and hard patterns as a characterization of weak-to-strong generalization. Charikar et al. (2024) studied a regression setting and characterized the gain in accuracy in weak-to-strong generalization as the misfit between the weak and strong model. Mulgund & Pabbaraju (2025) generalized the result of Charikar et al. (2024) to a general class of loss functions. Medvedev et al. (2025) considered a random feature

model with overspecified student and teacher models to establish weak-to-strong generalization and its quantitative limits. Somerstep et al. (2025) cast weak-to-strong generalization as a transfer learning problem where a latent concept is transferred from a weak model to a strong model. Wu & Sahai (2024) studied weak-to-strong generalization via benign overfitting under an overparameterized spiked covariance model. Ildiz et al. (2024) analyzed weak-to-strong generalization and scaling laws in the setting of high-dimensional linear regression with both model and distribution shifts. Their analysis leverages the equivalence between weak-to-strong generalization and covariate model shift to establish theoretical results. Dong et al. (2025) studied a ridgeless regression problem and established weak-to-strong generalization via the lens of intrinsic dimension. At a high level, our work, along with (Ildiz et al., 2024; Wu & Sahai, 2024), considers settings where the weak teacher is incapable of recovering the ground-truth function. However, in contrast to these works, we provide an explicit analysis of the student's learning dynamics, shedding light on how weak-to-strong generalization emerges over the course of training.

**Knowledge Distillation.** Weak-to-strong generalization is conceptually related to knowledge distillation, originally introduced by Buciluǎ et al. (2006); Hinton et al. (2015), in which a large, well-trained teacher model transfers knowledge to a smaller student model through soft or hard labels. A substantial body of theoretical work has investigated the mechanisms underlying knowledge distillation, including when and how the student benefits from the teacher's supervision (Phuong & Lampert, 2019; Allen-Zhu & Li, 2020; Ji & Zhu, 2020; Mobahi et al., 2020; Menon et al., 2020; Wei et al., 2020; Stanton et al., 2021; Nagarajan et al., 2023). In contrast, weak-to-strong generalization inverts this setup: a strong student is trained using supervision from a weak teacher.

**Empirical Studies of Weak to Strong Generalization.** Following the pioneering work of Burns et al. (2023), there is a growing body of work considering weak-to-strong generalization for different tasks, including LLM alignment (Ji et al., 2024; Tao & Li, 2024), vision superalignment (Guo et al., 2024), continual superalignment (Puthumanaillam et al., 2024), weak-to-strong deception (Yang et al., 2024a), and reasoning tasks (Yang et al., 2024b; Bansal et al., 2024). A recent survey by Kim et al. (2024) provides a comprehensive overview of these empirical developments.

# 3 PROBLEM SETUP AND PRELIMINARIES

**Notations.** Define $\| \cdot \|$ by the Euclidean norm. We define $M_{\mathsf{TE}}$, $M_{\mathsf{ST}}$, $d$ as the number of learnable parameters of the teacher, student and groudtruth, respectively. We use the asymptotic equivalent notation $u \simeq v$ for $u, v \in \mathbb{R}$, to mean that the ratio $\frac{u}{v}$ tends to 1 when the teacher model size $M_{\mathsf{TE}}$ and the groundtruth model size $d$ go to infinity. Similarly, we use asymptotic notation $u \gtrsim v$ and $u \lesssim v$ to denote $\frac{u}{v}$ tends to $+\infty$ and $0$ asymptotically. All of our asymptotic results for random quantities are understood to hold almost surely. We use the standard $O(\cdot), \Theta(\cdot), \Omega(\cdot)$ notations.

## 3.1 BACKGROUND AND PROBLEM SETUP

Define $f^* : \mathbb{R}^d \to \mathbb{R}$ as a groundtruth function, and $f : \mathbb{R}^d \to \mathbb{R}$ is a predictor. Then the population loss is defined as the mean squared error (MSE) between predictor $f$ and groundtruth $f^*$, i.e., $L(f) = \mathbb{E}_{x \sim \mathcal{D}} \left[ (f(x) - f^*(x))^2 \right]$, where $\mathcal{D}$ is the data distribution. The predictor $f$ refers to either the teacher or the student model in our paper. We consider the case where $f$ is modeled as a two-layer random feature model (Rahimi & Recht, 2007). A two-layer random feature model takes the form:

$$f(x) = \sigma(x^\top \mathbf{U})\boldsymbol{w}, \tag{1}$$

where $x \in \mathbb{R}^{d \times 1}$ is the input data, $\mathbf{U} \in \mathbb{R}^{d \times M}$ is a random matrix with $M$ being the number of hidden neurons, $\mathbf{w} \in \mathbb{R}^{M \times 1}$ is the last linear layer, and $\sigma(\cdot) : \mathbb{R}^{1 \times M} \to \mathbb{R}^{1 \times M}$ is the activation function taken element-wisely. In this paper, we fix the first layer weight $\mathbf{U}$ and only train the last layer weight $\boldsymbol{w}$.

**Teacher Model.** We consider the teacher model to be a two-layer neural network with random features. Similar to Medvedev et al. (2025), we assume the teacher is optimally trained and has access to the groundtruth distribution $\mathcal{D}$ and the groundtruth label. In particular, the optimally trained teacher model is defined as the minimizer of the least square loss

$$\boldsymbol{w}_{\mathsf{TE}} = \underset{\boldsymbol{w} \in \mathbb{R}^{M_{\mathsf{TE}}}}{\arg\min} \mathbb{E}_{x \sim \mathcal{D}} \left[ \|f^*(x) - \sigma(x^\top \mathbf{U}_{\mathsf{TE}})\boldsymbol{w}\|^2 \right], \tag{2}$$

where $\mathbf{U}_{\mathsf{TE}} \in \mathbb{R}^{d \times M_{\mathsf{TE}}}$ is a fixed random projection matrix, each element of $\mathbf{U}_{\mathsf{TE}}$ is independently sampled from $\mathcal{N}(0, M_{\mathsf{TE}}^{-1})$, and $\sigma(\cdot) : \mathbb{R}^{1 \times M_{\mathsf{TE}}} \to \mathbb{R}^{1 \times M_{\mathsf{TE}}}$ is the activation function. Therefore, the teacher predictor is defined as $f_{\mathsf{TE}}(x) = \sigma(x^\top \mathbf{U}_{\mathsf{TE}}) \boldsymbol{w}_{\mathsf{TE}}$.

**Student Model and Learning Dynamics.** The student model, also a two-layer neural network with random features, is trained using labels generated by the teacher. The learning dynamics of the student is governed by gradient flow with respect to the loss function $\mathbb{E}_{x \sim \mathcal{D}}[\|y_{\mathsf{TE}} - \sigma(x^\top \mathbf{U}_{\mathsf{ST}}) \boldsymbol{w}\|^2]$, where $\mathbf{U}_{\mathsf{ST}} \in \mathbb{R}^{d \times M_{\mathsf{ST}}}$ is a fixed random projection matrix, each element of $\mathbf{U}_{\mathsf{ST}}$ is independently sampled from $\mathcal{N}(0, M_{\mathsf{ST}}^{-1})$, $y_{\mathsf{TE}} = \sigma(x^\top \mathbf{U}_{\mathsf{TE}}) \boldsymbol{w}_{\mathsf{TE}}$, and $\sigma(\cdot) : \mathbb{R}^{1 \times M_{\mathsf{ST}}} \to \mathbb{R}^{1 \times M_{\mathsf{ST}}}$ is the activation function. In particular, the student learning dynamics is given by the following gradient flow,

$$\frac{d\boldsymbol{w}_{\mathsf{ST}}(t)}{dt} = -2\mathbb{E}_{x \sim \mathcal{D}} \left[ \left( y_{\mathsf{TE}} - \sigma(x^\top \mathbf{U}_{\mathsf{ST}}) \boldsymbol{w} \right) \cdot \sigma(\mathbf{U}_{\mathsf{ST}}^\top x) \right], \tag{3}$$

where $t$ is the training time. The student predictor is obtained by running the gradient flow dynamics (3) with a stopping time $T$, that is $f_{\mathsf{ST}}(x) = \sigma(x^\top \mathbf{U}_{\mathsf{ST}}) \boldsymbol{w}_{\mathsf{ST}}(T)$.

**Performance Measure for Weak-to-Strong Generalization.** Burns et al. (2023) proposed a performance measure for quantifying the weak-to-strong generalization namely **performance gap recovered** (PGR), which is defined as the fraction of the performance gap (the difference in performance between the weak and strong ceiling models) that we can recover with the weak supervision. In our setting, we follow Medvedev et al. (2025); Burns et al. (2023) to define the PGR as the following:

$$\mathrm{PGR} = \frac{\mathcal{E}_{\mathsf{TE}} - \mathcal{E}_{\mathsf{ST}}}{\mathcal{E}_{\mathsf{TE}} - \mathcal{E}_{\mathsf{ST}}^C} \geq \frac{\mathcal{E}_{\mathsf{TE}} - \mathcal{E}_{\mathsf{ST}}}{\mathcal{E}_{\mathsf{TE}}} = 1 - \frac{\mathcal{E}_{\mathsf{ST}}}{\mathcal{E}_{\mathsf{TE}}}, \tag{4}$$

where $\mathcal{E}_{\mathsf{TE}}$ and $\mathcal{E}_{\mathsf{ST}}$ denote the error of the teacher predictor and the student predictor respectively, and $\mathcal{E}_{\mathsf{ST}}^C$ is the ceiling performance of the student (when trained with groundtruth label and with access to the groundtruth data distribution). Therefore, a smaller MSE ratio between the student and teacher models, $\mathcal{E}_{\mathsf{ST}}/\mathcal{E}_{\mathsf{TE}}$, leads to a larger PGR, indicating stronger weak-to-strong generalization.

In this paper, we focus on the setting where the teacher is underspecified and the student is overspecified: $M_{\mathsf{ST}} \gg d > M_{\mathsf{TE}}$.

# 4 THEORETICAL CHARACTERIZATION OF WEAK TO STRONG GENERALIZATION

In this section, we provide a theoretical characterization of weak-to-strong generalization, where both the teacher and student are modeled as two-layer neural networks. We consider linear neural networks in Section 4.1 and ReLU networks in Section 4.2, respectively. The proof sketch is included in Section 4.3 and the rest of the proofs can be found in Appendix A and B.

## 4.1 LINEAR GROUNDTRUTH AND TWO-LAYER LINEAR NETWORKS

In this subsection, we present the theoretical results under the following linear groundtruth model, and the identity activation function, i.e., $\sigma(a) = a$ in learning model (1).

**Groundtruth Model.** The groundtruth function is a linear function: $f^*(x) = x^\top \boldsymbol{\theta}^*$, where $x \sim \mathcal{N}(0, \Lambda)$, where $\Lambda = \mathrm{diag}\{\lambda_1, \ldots, \lambda_d\} \in \mathbb{R}^{d \times d}$ with $\lambda_1 \geq \lambda_2 \geq \cdots \geq \lambda_d > 0$ and $\boldsymbol{\theta}^* \in \mathbb{R}^d$. Note that $M_{\mathsf{ST}} \gg d > M_{\mathsf{TE}}$. Hence, the $i$-th eigen direction is $\boldsymbol{e}_i = (0, \ldots, 1, \ldots, 0)^\top \in \mathbb{R}^d$ (with 1 in the $i$-th position), i.e., the $i$-th standard basis vector in $\mathbb{R}^d$.

Denote $\mathbf{P}_{\mathsf{TE}} = \Lambda^{1/2} \mathbf{U}_{\mathsf{TE}} (\mathbf{U}_{\mathsf{TE}}^\top \Lambda \mathbf{U}_{\mathsf{TE}})^{-1} \mathbf{U}_{\mathsf{TE}}^\top \Lambda^{1/2}$ as the teacher model's learning transfer matrix (Simon et al., 2023). Denote the teacher's MSE as $\mathcal{E}_{\mathsf{TE}} = \mathbb{E}[(f^*(x) - x^\top \mathbf{U}_{\mathsf{TE}} \boldsymbol{w}_{\mathsf{TE}})^2]$, and denote the student MSE at time $t$ as $\mathcal{E}_{\mathsf{ST}}(t) = \mathbb{E}[(f^*(x) - x^\top \mathbf{U}_{\mathsf{ST}} \boldsymbol{w}_{\mathsf{ST}}(t))^2]$. Note that $\mathbb{E}(\cdot)$ takes total expectation over randomness in data and model parameters of the first layer. Let $\kappa_{\mathsf{TE}}$ be unique root of the equation $\sum_{i=1}^d \lambda_i / (\lambda_i + \kappa_{\mathsf{TE}}) = M_{\mathsf{TE}}$, which characterizes the learnability of the model (according to Theorem 4.1 in LeJeune et al. (2024)). We also define the *expected signal* vector (Janson et al., 2017) as $v^* = \Lambda^{1/2} \boldsymbol{\theta}^* = (\sqrt{\lambda_1} \theta_1^*, \ldots, \sqrt{\lambda_d} \theta_d^*)^\top$.

Our first result is the asymptotic characterization of the MSE of the teacher model defined in (2).

**Theorem 4.1** (Teacher MSE of linear model). *Under the linear ground truth function, denote* $L_i = \frac{\lambda_i}{\lambda_i + \kappa_{\mathsf{TE}}}$ *for* $i \in [d]$. *The MSE of the teacher model is given as*

$$\mathcal{E}_{\mathsf{TE}} \simeq \frac{M_{\mathsf{TE}}}{M_{\mathsf{TE}} - \sum_{i \in [d]} L_i^2} \cdot \sum_{i \in [d]} (1 - L_i)^2 (v_i^*)^2. \tag{5}$$

**Remark 1:** Note that $L_i$, referred to as the *eigenmode learnability* (Simon et al., 2023), quantifies how well the teacher model captures the $i$-th eigenmode of the data covariance matrix $\Lambda$. By definition, $L_i \in [0, 1]$ and $\sum_{i=1}^{d} L_i = M_{\mathsf{TE}}$. Consequently, when the teacher is underspecified (i.e., $M_{\mathsf{TE}} < d$), there must exist some directions $e_i$ for which $L_i$ is significantly smaller than 1, indicating that the teacher fails to fully learn the corresponding components of the ground truth.

**Remark 2:** By analyzing the teacher error expression in (5), we can show that the teacher's MSE can become large when the eigenvalue distribution of the data covariance matrix $\Lambda$ is clustered–specifically, when the spectrum consists of two groups: one with large eigenvalues and another with small ones. In this case, the eigenmode learnabilities $L_i$ tend to concentrate near either 1 or 0. Consequently, we have $\sum_{i=1}^{d} L_i^2 \approx \sum_{i=1}^{d} L_i = M_{\mathsf{TE}}$, which leads to a large teacher error in (5) whenever there is significant expected signal in directions corresponding to small eigenvalues–i.e., when there exists some $i$ such that $\lambda_i \to 0$ and $|\theta_i^*|$ is large enough.

**Theorem 4.2** (Student MSE of linear model). *Under the linear ground truth function, as* $M_{\mathsf{ST}} \to \infty$, *the MSE of the student model at time* $t$ *can be represented as*

$$\mathcal{E}_{\mathsf{st}}(t) \simeq \sum_{i \in [d]} A_i \left( \gamma_i(t) - \frac{B_i}{A_i} \right)^2 + (v_i^*)^2 - \frac{B_i^2}{A_i}, \tag{6}$$

*where* $\gamma_i(t) = (1 - e^{-\lambda_i t}) L_i$, $A_i = \frac{L_i^2 \mathcal{E}_{\mathsf{TE}}}{M_{\mathsf{TE}}} + L_i^2 (v_i^*)^2$, *and* $B_i = L_i (v_i^*)^2$.

**Remark 3:** From the learning dynamics described in (3), Theorem 4.2 established how the student MSE changes over time. In particular, we observe that the error in each eigenvector direction is a quadratic function on $\gamma_i(t)$, which indicates the importance of early stopping in the training process of the student model. Notably, the error in the direction with a larger eigenvalue converges to its minimum error rapidly, with the resulting error being significantly smaller than that of the teacher model in the same direction. If the ground truth signals are predominantly concentrated in these directions, we can show that the student model trained with early stopping achieves better generalization than the teacher model.

Next, we provide a specific example of the linear ground truth function, where the MSE ratio asymptotically tends to zero, i.e., the weak-to-strong generalization provably occurs with PGR goes to 1 asymptotically.

**Theorem 4.3** (Upper bound of MSE ratio). *Under the linear ground truth function, we let* $M_{\mathsf{TE}} = \Theta(m)$, $d = \Theta(m^{3/2})$ *when* $m \to \infty$. *Given* $K = \Theta(m^{1/2})$, *we set the covariance matrix as* $\lambda_i = \Theta(1)$ *if* $i \leq K$, *and* $\lambda_i = \Theta(m^{-1})$ *for* $K < i \leq d$, *and choose the ground truth weight such that* $\lambda_i(\theta_i^*)^2 = \Theta(m^\alpha)$ *if* $i \leq K$ *and* $\lambda_i(\theta_i^*)^2 = \Theta(m^{-\beta})$. *If* $\alpha \geq 1$, $\beta > 0$ *and* $\alpha + \beta \in (2, 3)$, *when* $T = \frac{1}{2} \log \left( \frac{\mathcal{E}_{\mathsf{TE}} + M_{\mathsf{TE}}}{\mathcal{E}_{\mathsf{TE}} - \kappa_{\mathsf{TE}} M_{\mathsf{TE}}} \right) = O(\log m)$, *the MSE ratio satisfies*

$$\mathcal{E}_{\mathsf{ST}}(T) / \mathcal{E}_{\mathsf{TE}} \lesssim \Theta \left( M_{\mathsf{TE}}^{-1/2} \right) + \Theta \left( M_{\mathsf{TE}}^{-(\alpha+\beta)+2} \right). \tag{7}$$

**Remark 4:** When $\alpha > 1$, the signal is concentrated along high-variance directions of the data covariance matrix, which is essential for the teacher to capture meaningful information that can be effectively transferred to the student. An additional key condition is that $\alpha + \beta \in (2, 3)$, which implicitly imposes both upper and lower bounds on $\beta$ such that the data covariance matrix is not severely ill-conditioned. This ensure that the teacher loss stays in a reasonable range to enable weak-to-strong generalization with PGR going to 1 when $M_{TE}$ goes to $+\infty$.

By using the quadratic property of student MSE in (6), we can derive the limit of weak-to-strong generalization for the linear ground truth function.

**Theorem 4.4** (Lower bound of MSE ratio). *For linear models, given arbitrary covariance matrix $\Lambda$, non-zero ground truth $\boldsymbol{\theta}^*$, the following lower bound holds for any $T \geq 0$: $\mathcal{E}_{\mathsf{ST}}(T)/\mathcal{E}_{\mathsf{TE}} \gtrsim \sum_{i \in [d]} \left( \mathcal{E}_{\mathsf{TE}}/(v_i^*)^2 + M_{\mathsf{TE}} \right)^{-1}$.*

**Remark 5:** Fixing $\Lambda$ and $\boldsymbol{\theta}^*$, the lower bound of the MSE ratio is determined by the teacher error $\mathcal{E}_{\mathsf{TE}}$ and teacher model size. Together with the upper bound in (7), the MSE-ratio becomes smaller if we increase the teacher model size.

## 4.2 Nonlinear Groundtruth and Two-Layer ReLU Networks

In this subsection, we present the theoretical results under the following ReLU groundtruth model, i.e., $\sigma(a) = \max\{a, 0\}$ in the learning model.

**Groundtruth Model.** The groundtruth function $f^*$ is an even polynomial of degree $k^*$. Let $\{e_i(\cdot)\}_{i \geq 1}^{\infty}$ be an orthonormal spherical–harmonic basis grouped by degree $k \in \{0, 1, 2, \dots\}$ with multiplicities $N_k = \frac{(2k+d-2)(k+d-3)!}{k! \, (d-2)!}$. By Proposition B.3 in Medvedev et al. (2025) (see Lemma C.4), we have $f^*(x) = \sum_{i=1}^{M^*} v_i^* e_i(x)$, where $M^* = \sum_{j=1}^{k^*} N_j$ and $v_i^* := \sqrt{\lambda_i} w_i^*$.

We assume that the teacher model $f_{\mathsf{TE}}$ and the student model $f_{\mathsf{st}}$ are even polynomials of degree $k_{\mathsf{TE}}$ and $k_{\mathsf{st}}$, respectively. Denote $M_{\mathsf{TE}} := \sum_{j=0}^{k_{\mathsf{TE}}} N_j$ and $M_{\mathsf{ST}} := \sum_{j=0}^{k_{\mathsf{st}}} N_j$. Define the feature–covariance matrix as $\Lambda = \mathrm{diag}\{\underbrace{\sigma_1, \dots, \sigma_1}_{N_1 \text{ times}}, \underbrace{\sigma_2, \dots, \sigma_2}_{N_2 \text{ times}}, \dots, \underbrace{\sigma_k, \dots, \sigma_k}_{N_k \text{ times}}\} := \mathrm{diag}(\lambda_1, \dots, \lambda_{M^*})$, Let $\kappa_{\mathsf{TE}}$ be unique root of the equation $\sum_{i=1}^{M^*} \lambda_i/(\lambda_i + \kappa_{\mathsf{TE}}) = M_{\mathsf{TE}}$. We also define the *expected signal* vectors as $v^* := (\sqrt{\lambda_1} w_1^*, \dots, \sqrt{\lambda_{M^*}} w_{M^*}^*)$ and the teacher prediction signal vectors as $v_{\mathsf{TE}} = (v_{\mathsf{TE},1}, \dots, v_{\mathsf{TE},M^*}) \in \mathbb{R}^{M^*}$. Mark $\mathrm{Unif}(\mathbb{S}^{d-1})$ as $\mathcal{D}$.

**Theorem 4.5.** *Under the ReLU ground truth function, and the Gaussian Universality assumption over random features (Simon et al., 2023; Medvedev et al., 2025), denote $L_i = \frac{\lambda_i}{\lambda_i + \kappa_{\mathsf{TE}}}$, $i \in [M^*]$. (1) The MSE of the teacher model satisfies $\mathcal{E}_{\mathsf{TE}} \simeq \frac{M_{\mathsf{TE}}}{M_{\mathsf{TE}} - \sum_{j \in [M^*]} L_j^2} \sum_{i \in [M^*]} (1 - L_i)^2 (v_i^*)^2$. (2) As $M_{\mathsf{ST}} \to \infty$, the MSE of the student model satisfies $\mathcal{E}_{\mathsf{st}}(t) \simeq \sum_{i=1}^{M^*} A_i (\gamma_i(t) - B_i/A_i)^2 + (v_i^*)^2 - \frac{B_i^2}{A_i}$, where $\gamma_i(t) = (1 - e^{-\lambda_i t}) L_i$, $A_i = \frac{L_i^2 \mathcal{E}_{\mathsf{TE}}}{M_{\mathsf{TE}}} + L_i^2 (v_i^*)^2$, and $B_i = L_i (v_i^*)^2$.*

*In addition, for arbitrary non-zero $f^*$ and any $T \geq 0$, we also have $\mathcal{E}_{\mathsf{st}}(T)/\mathcal{E}_{\mathsf{TE}} \gtrsim \sum_{i \in [M^*]} (\mathcal{E}_{\mathsf{TE}}/(v_i^*)^2 + M_{\mathsf{TE}})^{-1}$.*

**Remark 6:** Though the linear setting and ReLU setting have the same MSE formula for both the teacher and the student, the proof techniques are different. Unlike generic linear model, a fixed–first-layer ReLU random-feature model behaves like a linear setting only in the spherical-harmonic basis, where the feature covariance is diagonal with a certain degree-wise eigenspectrum.

Accordingly, we (i) characterize the eigenvalues and their multiplicities in harmonic coordinates (see the proof of Lemma B.1), (ii) derive the projection operator and its coordinate-wise variance (following the proof of Lemma A.1), and (iii) characterize the teacher prediction and its test risk (see Theorem 4.5).

**Theorem 4.6.** *Consider the groundtruth setting: let $r \in (1, 3), \alpha > 1, \beta < \alpha - 1/2$ and $k_{st} = k^* = k_{te}^r$, $k_{te} \gg d = 4$, $w_1^* = m^\alpha$, $w_i^* = 0$ for $2 \leq i \leq N_{k^*-1}$, and $w_j^* = m^\beta$ for $N_{k^*-1} < j \leq N_{k^*}$, where $m := M_{\mathsf{TE}} = \sum_{k=0}^{k_{te}} N_k = \Theta(k_{te}^{d-1})$ and $M^* = m^r = \Theta(k^{*(d-1)})$. Then weak-to-strong ratio satisfies $\mathcal{E}_{\mathsf{st}}(T)/\mathcal{E}_{\mathsf{TE}} \leq \Theta(m^{-1}) + \Theta(m^{2\beta - \frac{r}{3} - 2\alpha + 2})$.*

**Remark 7:** This setting aligns the ground-truth expansion and the eigenvalue ordering with the ReLU kernel's spherical–harmonic decomposition.

## 4.3 Proof Sketch

In this section, we outline the proof strategy for Theorem 4.1 and 4.2, and that for Theorem 4.5 is similar. Our analysis is mainly based on the eigenlearning framework (Simon et al., 2023). Note that the proof techniques in Section 4.3 apply to both linear and ReLU ground-truth models.

**Lemma 4.7** (Representation of teacher MSE). *Denote the projection matrix of teacher model as* $\mathbf{P}_{\mathsf{TE}} = \Lambda^{1/2}\mathbf{U}_{\mathsf{TE}}\left(\mathbf{U}_{\mathsf{TE}}^{\top}\Lambda\mathbf{U}_{\mathsf{TE}}\right)^{-1}\mathbf{U}_{\mathsf{TE}}^{\top}\Lambda^{1/2}$ *and the ground truth signal vector as* $v^* = \Lambda^{1/2}\theta^*$. *The conditional MSE of the teacher model can be represented as* $\mathbb{E}_{x\sim\mathcal{D}}[(y - x^{\top}\mathbf{U}_{\mathsf{TE}}w)^2] = \|v^* - v_{\mathsf{TE}}\|^2$, *where* $v_{\mathsf{TE}} = \mathbf{P}_{\mathsf{TE}}v^*$ *is the signal vector learned by teacher model.*

The lemma above shows that the MSE of the teacher model is identical to the projection error of the expected signal vector $v^*$. From a high level, the projected vector $v_{\mathsf{TE}}$ is what the teacher model learns from the data population. By taking full expectation, and leveraging random matrix theory to analyze the asymptotic behavior of the projection matrix $\mathbf{P}_{\mathsf{TE}}$, we can prove Theorem 4.1. Since the student model is trained using the teacher model's predicted label, the next lemma represents the training loss function as the distance between $v_{\mathsf{TE}}$ and the signal captured by the student model.

**Lemma 4.8** (Representation of student training loss). *Given the weight* $\boldsymbol{w}_{\mathsf{ST}}(t)$*, the loss of the student model can be represented as* $\mathbb{E}_{x\sim\mathcal{D}}[(y_{\mathsf{TE}} - x^{\top}\mathbf{U}_{\mathsf{ST}}\boldsymbol{w})^2] = \|v_{\mathsf{TE}} - \Lambda^{1/2}\mathbf{U}_{\mathsf{ST}}\boldsymbol{w}_{\mathsf{ST}}(t)\|^2$.

With the loss representation in Lemma 4.8, we can equivalently write the gradient dynamics as $d\boldsymbol{w}_{\mathsf{ST}}(t)/dt = -2\mathbf{U}_{\mathsf{ST}}^{\top}\Lambda^{1/2}\left(v_{\mathsf{ST}}(t) - v_{\mathsf{TE}}\right)$. Let $v_{\mathsf{ST}}(t) = \Lambda^{1/2}\mathbf{U}_{\mathsf{ST}}\boldsymbol{w}_{\mathsf{ST}}(t)$ be the student signal vector. Under the definition of random projection matrix $\mathbf{U}_{\mathsf{ST}}$, according to Bai-Yin's law (Bai et al., 1993), we can guarantee $\mathbf{U}_{\mathsf{ST}}\mathbf{U}_{\mathsf{ST}}^{\top} \simeq \mathbf{I}_d$ when $M_{\mathsf{ST}} \to \infty$. Then we can have the dynamics of the expected signal vector as $\frac{d}{dt}v_{\mathsf{ST}}(t) \simeq -2\sum_{i=1}^{d}\lambda_i\langle v_{\mathsf{ST}}(t) - v_{\mathsf{TE}}, \boldsymbol{e}_i\rangle \cdot \boldsymbol{e}_i$. Consider the following ODE with initializing at $v_{\mathsf{ST}}(0) = 0$: $\frac{d}{dt}\langle v_{\mathsf{ST}}(t) - v_{\mathsf{TE}}, \boldsymbol{e}_i\rangle = -2\lambda_i\langle v_{\mathsf{ST}}(t) - v_{\mathsf{TE}}, \boldsymbol{e}_i\rangle$, we can obtain the solution as $\langle v_{\mathsf{ST}}(t) - v_{\mathsf{TE}}, \boldsymbol{e}_i\rangle = e^{-2\lambda_i t}\langle v_{\mathsf{ST}}(0) - v_{\mathsf{TE}}, \boldsymbol{e}_i\rangle$, which gives a precise characterization of the student dynamics stated in the following lemma.

**Lemma 4.9** (Dynamics of student signal). *Denote the signal vector of student model as* $v_{\mathsf{ST}}(t) = \Lambda^{1/2}\mathbf{U}_{\mathsf{ST}}\boldsymbol{w}_{\mathsf{ST}}(t)$. *The time-dependent student signal is given by* $v_{\mathsf{ST}}(t) \simeq \boldsymbol{\gamma}(t) \odot v_{\mathsf{TE}}$, *where* $\boldsymbol{\gamma}(t) := (\gamma_1(t), \ldots, \gamma_d(t))^{\top}$.

## 5 EXPERIMENTS

In this section, we provide experiments to verify our theory and explore directions for future research. First, in Section 5.1 we present experiments using exactly the data distribution and predictors from Theorem 4.3, in order to numerically confirm our theoretical findings. Next, we depart from our theoretical setting to understand when weak-to-strong generalization can occur in practical contexts: in Section 5.2, we explore the effect of varying the sizes of the student and teacher; in Section 5.3, we use an alternate data model in which the teacher, student, and groundtruth function are two-layer ReLU networks. We also conducted experiments on real-world data, which is included in Appendix D. All experiments were run on a single NVIDIA RTX 3090.

### 5.1 SIMULATING THEOREM 4.3

**Setup.** Since Theorem 4.3 specifies a data distribution and predictors, we can faithfully simulate the theoretical setting up to an approximation of the student and teacher parameters: to approximate the optimally trained teacher (Equation 2) and the gradient flow dynamics of the student parameter (Equation 3), we train the teacher and the student by stochastic gradient descent (SGD) with online data, using a learning rate of $0.1$ and a batch size of $1024$. Both the teacher and the student are trained for 1000 SGD steps. All other details are exactly as specified in Theorem 4.3, where we chose $\alpha = 1.5, \beta = 1.0$. All multiplicative constants omitted by $\Theta(\cdot)$ in the theorem statement are set to 1. For each choice of $M_{\mathsf{TE}}$, we run a single random seed.

**Results.** Figure 1 shows the loss ratio $\mathcal{E}_{\mathsf{ST}}(t)/\mathcal{E}_{\mathsf{TE}}$ as $M_{\mathsf{TE}}$ varies over $\{2^2, 2^3, \ldots, 2^7\}$, together with the teacher and student losses for a single case $M_{\mathsf{TE}} = 64$. From Figure 1(a), we can see for each value of $M_{\mathsf{TE}}$, the loss ratio initially decreases below $1.0$, then slowly increases, eventually reaching $1.0$ or above, so that early stopping of the student is indeed crucial for achieving weak-to-strong generalization. We can see in Figure 1(b) that the student training loss (with teacher-generated labels) goes to zero over time, and simultaneously the student's population loss approaches that of the teacher if the student is trained for a long time. Further, as predicted by our theory, the minimum value of the loss ratio over time decreases with $M_{\mathsf{TE}}$, getting closer to 0 as $M_{\mathsf{TE}}$ increases.

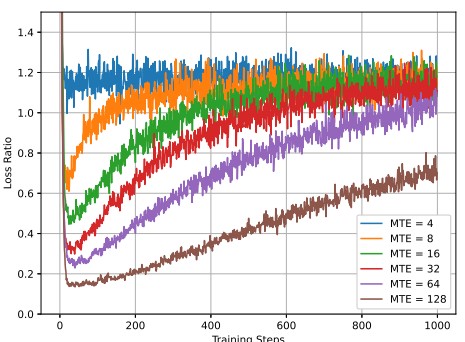 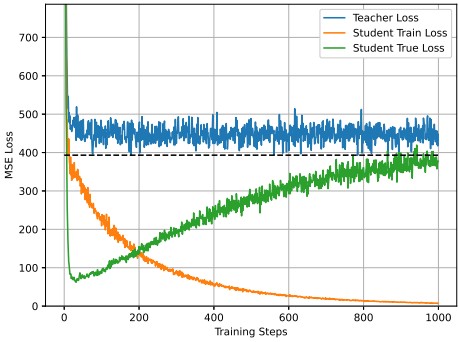

(a) Loss ratio $\mathcal{E}_{\mathsf{ST}}(t)/\mathcal{E}_{\mathsf{TE}}$ with varying $M_{\mathsf{TE}}$.  (b) Teacher and student loss with $M_{\mathsf{TE}} = 64$.

Figure 1: Simulation of setting from Theorem 4.3. Left: Ratio of student loss over time vs. smallest teacher loss. Right: Teacher loss, student training loss (with teacher-generated labels), and student loss (with true labels) for the case $M_{\mathsf{TE}} = 64$. Smallest teacher loss is shown with a dashed line. Values of $d$, $M_{\mathsf{ST}}$ corresponding to each value of $M_{\mathsf{TE}}$ are given in Table 2 in Appendix D.

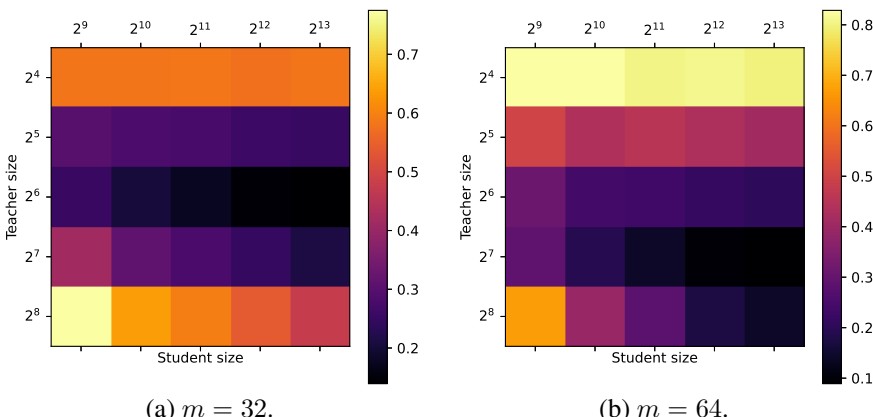

(a) $m = 32$.  (b) $m = 64$.

Figure 2: Ratio of student loss to teacher loss for various teacher sizes $M_{\mathsf{TE}}$ and student sizes $M_{\mathsf{ST}}$, using the data distribution and ground-truth predictor of Theorem 4.3.

## 5.2 Effect of Teacher vs. Student Size

**Setup.** We observe how the relative sizes of the teacher $M_{\mathsf{TE}}$ and the student $M_{\mathsf{ST}}$ affect weak-to-strong generalization under the setting of Theorem 4.3. In this experiment, we use the data distribution and ground-truth predictor specified in Theorem 4.3 in terms of $m$, but we allow $M_{\mathsf{TE}}$ and $M_{\mathsf{ST}}$ to vary independently of $m$. Besides this, we follow all settings from Section 5.1. We evaluate every pair $(M_{\mathsf{TE}}, M_{\mathsf{ST}})$ with $M_{\mathsf{TE}} \in \{2^4, 2^5, \ldots, 2^8\}$ and $M_{\mathsf{ST}} \in \{2^9, 2^{10}, \ldots, 2^{13}\}$, and for each pair, we compute the ratio of the smallest student loss and the smallest teacher loss over time. For each pair, we run a single random seed.

**Results.** Figure 2 shows the ratio of student loss to teacher loss for every pair $(M_{\mathsf{TE}}, M_{\mathsf{ST}})$, for two values of the data distribution parameter $m \in \{32, 64\}$. We see that the occurrence of weak-to-strong generalization is very reliable in this setting, since the student loss is smaller than the teacher loss for every combination of network sizes. However, there is significant variation in the size of this ratio: for a fixed teacher size, the ratio tends to decrease as the student size increases, and for a fixed student size, the ratio is smallest for moderate teacher sizes. These results are not entirely consistent with those of Burns et al. (2023), who found that a smaller ratio of student size to teacher size leads to stronger weak-to-strong generalization, although they used real-world data while we used synthetic data. This result may stem from the simplified data model adopted in Theorem 4.3,

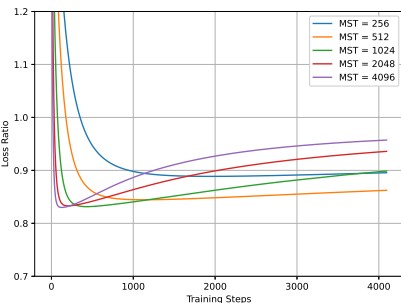 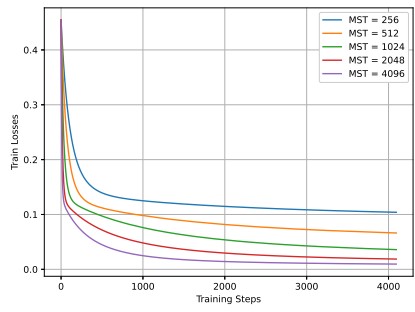

(a) Loss ratio $\mathcal{E}_{\mathsf{ST}}(t)/\mathcal{E}_{\mathsf{TE}}$ with varying $M_{\mathsf{ST}}$.     (b) Training loss of student (under teacher label).

Figure 3: Weak-to-strong generalization in two-layer ReLU networks with offline data. Left: Ratio of student test loss over time versus smallest teacher test loss. Right: Student train loss with teacher-generated labels. Ground-truth hidden size is $d = 64$ and teacher hidden size is $M_{\mathsf{TE}} = 48$.

which captures certain aspects of weak-to-strong generalization—such as those demonstrated in our simulation experiments (Section 5.1)—but fails to account for more complex behaviors, such as those highlighted in **(E3)** in Section 1, where the sizes of the teacher and student models can vary independently of the ground-truth model in practice. It remains open to theoretically characterize the effect of teacher/student sizes on weak-to-strong generalization, which we leave to future work.

## 5.3 ReLU Networks

**Setup.** Now, we step outside of the data model prescribed by our theory to determine whether weak-to-strong generalization can be empirically observed with a more practical setup. To do so, we use two-layer ReLU networks for the ground-truth, student, teacher. Here we fix the hidden size of the ground-truth model $d = 64$ and the teacher $M_{\mathsf{TE}} = 48$, while allowing the student's hidden size to vary $M_{\mathsf{ST}} \in \{2^8, 2^9, \ldots, 2^{12}\}$. Consistently with Section 3.1, only the second layer weights of the student and teacher are trained. Different from Section 5.1, here we use offline data as a more realistic alternative to directly training on the population data: we train on a fixed dataset of $n = 2048$ samples, then evaluate on a held-out set of another $n$ samples. The student and teacher are trained for 4096 iterations of full-batch GD with learning rate 0.003 for the teacher and 0.001 for the student. After sampling data, the training process is fully deterministic.

**Results.** Figure 3 shows the ratio $\mathcal{E}_{\mathsf{ST}}(t)/\mathcal{E}_{\mathsf{TE}}$ between the student loss over time versus the smallest teacher loss encountered through training. Note that for this setting, the same teacher is used to generate labels for all students. From Figure 3, we make several observations. (1) Weak-to-strong generalization does occur in ReLU networks with offline data, which is supported by the fact that every student network reaches a loss ratio smaller than 1.0. This suggests that our theoretical findings may extend beyond the settings investigated in Section 4. (2) Early stopping is necessary to achieve the smallest loss ratio, although, differently from Figure 1, the loss ratio plateaus below 1.0 even when the number of iterations is large. (3) Larger students exhibit a slightly greater degree of weak-to-strong generalization, overfit faster, and achieve a larger final loss ratio. Overall, these results suggest that weak-to-strong generalization can occur beyond our currently established theory, and we leave the theoretical analysis of such settings (ReLU networks, offline data) for future work.

## 6 Conclusion

In this paper, we are motivated by the practical observation that weak-to-strong generalization often arises in settings where the teacher model, trained on ground-truth labels, exhibits persistently high loss. To understand this phenomenon, we investigate its underlying mechanisms in a controlled theoretical setting involving random feature student and teacher models. Specifically, we consider a linear ground-truth model, an underspecified teacher, and an overspecified student. We analyze the teacher's intrinsic error due to limited capacity and identify conditions under which weak-to-strong generalization provably occurs, characterized by spectral properties of the data covariance matrix. We further explore the scenario of the ground-truth function represented by a two-layer ReLU network. Our theoretical findings are supported by simulations and real-world data.

REPRODUCIBILITY STATEMENT

We provide complete proofs of Theorems 4.1 to 4.4 in Section A, and the proof of Theorems 4.5 and 4.6 in Section B.

An anonymized code archive is included in the supplementary materials, containing training and evaluation scripts, configuration files, random seeds, and environment specifications. All experiments are conducted on the MNIST dataset, which is open source and distributed under the MIT license. We include preprocessing and splitting scripts.

Together, these materials are sufficient to fully reproduce our theoretical and empirical results.

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

## A  DEFERRED PROOFS OF LINEAR MODELS

### A.1  TEACHER MODEL

#### A.1.1  PROOF OF LEMMA 4.7

*Proof of Lemma 4.7.* According to the definition of $\mathcal{D}$, we know $\mathbb{E}_{x\sim\mathcal{D}}[xx^\top] = \Lambda$ and

$$w_{\mathsf{TE}} = \left(\mathbf{U}_{\mathsf{TE}}^\top\mathbb{E}_{x\sim\mathcal{D}}\left[xx^\top\right]\mathbf{U}_{\mathsf{TE}}\right)^{-1}\mathbf{U}_{\mathsf{TE}}^\top\mathbb{E}_{x\sim\mathcal{D}}\left[xy\right] = \left(\mathbf{U}_{\mathsf{TE}}^\top\Lambda\mathbf{U}_{\mathsf{TE}}\right)^{-1}\mathbf{U}_{\mathsf{TE}}^\top\Lambda\theta^*. \tag{8}$$

Then the MSE of the teacher model's prediction can be represented as

$$\mathbb{E}_{x\sim\mathcal{D}}\left[\left(y_{\mathsf{TE}} - x^\top\mathbf{U}_{\mathsf{TE}}w_{\mathsf{TE}}\right)^2\right]$$

$$= \mathbb{E}_{x\sim\mathcal{D}}\left[\left(x^\top\left(\mathbf{I}_d - \mathbf{U}_{\mathsf{TE}}\left(\mathbf{U}_{\mathsf{TE}}^\top\Lambda\mathbf{U}_{\mathsf{TE}}\right)^{-1}\mathbf{U}_{\mathsf{TE}}^\top\Lambda\right)\theta^*\right)^2\right]$$

$$= (\theta^*)^\top\left(\mathbf{I}_d - \mathbf{U}_{\mathsf{TE}}\left(\mathbf{U}_{\mathsf{TE}}^\top\Lambda\mathbf{U}_{\mathsf{TE}}\right)^{-1}\mathbf{U}_{\mathsf{TE}}^\top\Lambda\right)^\top\mathbb{E}_{x\sim\mathcal{D}}[xx^\top]\left(\mathbf{I}_d - \mathbf{U}_{\mathsf{TE}}\left(\mathbf{U}_{\mathsf{TE}}^\top\Lambda\mathbf{U}_{\mathsf{TE}}\right)^{-1}\mathbf{U}_{\mathsf{TE}}^\top\Lambda\right)\theta^*$$

$$= (\theta^*)^\top\Lambda\theta^* - (\theta^*)^\top\Lambda\mathbf{U}_{\mathsf{TE}}\left(\mathbf{U}_{\mathsf{TE}}^\top\Lambda\mathbf{U}_{\mathsf{TE}}\right)^{-1}\mathbf{U}_{\mathsf{TE}}^\top\Lambda\theta^*$$

$$= (\Lambda^{1/2}\theta^*)^\top\Lambda^{1/2}\theta^* - (\Lambda^{1/2}\theta^*)^\top\mathbf{P}_{\mathsf{TE}}(\Lambda^{1/2}\theta^*), \tag{9}$$

where

$$\mathbf{P}_{\mathsf{TE}} = \Lambda^{1/2}\mathbf{U}_{\mathsf{TE}}\left(\mathbf{U}_{\mathsf{TE}}^\top\Lambda\mathbf{U}_{\mathsf{TE}}\right)^{-1}\mathbf{U}_{\mathsf{TE}}^\top\Lambda^{1/2}. \tag{10}$$

Denote $v^* = \Lambda^{1/2}\theta^* = (\sqrt{\lambda_1}\theta_1^*, \ldots, \sqrt{\lambda_d}\theta_d^*)^\top =: (v_1^*, \ldots, v_d^*)^\top$ as the signal vector. Since $\mathbf{P}_{\mathsf{TE}}$ is a projection matrix, we can also write MSE of the teacher model as

$$\mathbb{E}_{x\sim\mathcal{D}}\left[\left(y - x^\top\mathbf{U}_{\mathsf{TE}}w_{\mathsf{TE}}\right)^2\right] = \|v^* - \mathbf{P}_{\mathsf{TE}}v^*\|^2. \tag{11}$$

$\square$

#### A.1.2  PROOF OF THEOREM 4.1

*Proof of Theorem 4.1.* Using (14) in Lemma A.1, we have

$$\mathbb{E}\left[\mathbf{P}_{\mathsf{TE}}v^*\right] \simeq \mathrm{diag}\left\{\frac{\lambda_1}{\lambda_1 + \kappa_{\mathsf{TE}}}, \ldots, \frac{\lambda_d}{\lambda_d + \kappa_{\mathsf{TE}}}\right\}v^* = (L_1 v_1, \ldots, L_d v_d). \tag{12}$$

Applying (15) in Lemma A.1, we can compute the variance as

$$\mathrm{Var}[(\mathbf{P}_{\mathsf{TE}}v^*)_i] = 2\sum_{k,l\in[d]}\mathrm{Cov}[(\mathbf{P}_{\mathsf{TE}})_{ik}, (\mathbf{P}_{\mathsf{TE}})_{il}]v_k^* v_l^*$$

$$= \sum_{k,l\in[d]}v_k^* v_l^* \frac{L_i^2(1 - L_k)(1 - L_l)}{M_{\mathsf{TE}} - \sum_{i\in[d]}L_i^2}\mathbf{1}_{kl}$$

$$\simeq \frac{L_i^2}{M_{\mathsf{TE}} - \sum_{i\in[d]}L_i^2}\sum_{j\in[d]}(1 - L_j)^2(v_j^*)^2. \tag{13}$$

Combining the expectation and variance relations above, we have

$$\mathcal{E}_{\mathsf{TE}} = \mathbb{E}\left[\|v^* - \mathbf{P}_{\mathsf{TE}}v^*\|^2\right]$$

$$= \sum_{i\in[d]}(v_i^*)^2 - 2v_i\mathbb{E}[(\mathbf{P}_{\mathsf{TE}}v^*)_i] + \mathbb{E}[(\mathbf{P}_{\mathsf{TE}}v^*)_i^2]$$

$$= \sum_{i\in[d]}(v_i^*)^2 - 2v_i\mathbb{E}[(\mathbf{P}_{\mathsf{TE}}v^*)_i] + (\mathbb{E}[(\mathbf{P}_{\mathsf{TE}}v^*)_i])^2 + \mathrm{Var}[(\mathbf{P}_{\mathsf{TE}}v^*)_i])$$

$$\simeq \sum_{i\in[d]}(1 - L_i)^2(v_i^*)^2 + \frac{L_i^2}{M_{\mathsf{TE}} - \sum_{j\in[d]}L_j^2}\sum_{j\in[d]}(1 - L_j)^2(v_j^*)^2$$

$$= \frac{M_{\mathsf{TE}}}{M_{\mathsf{TE}} - \sum_{j\in[d]}L_j^2}\sum_{i\in[d]}(1 - L_i)^2(v_i^*)^2.$$

$\square$

### A.1.3 PROOF OF LEMMA A.1

**Lemma A.1.** *Let $\kappa_{\mathsf{TE}}$ be the root of equation $\sum_{i=1}^{d} \frac{\lambda_i}{\lambda_i + \kappa} = M_{\mathsf{TE}}$ for $\kappa \in \mathbb{R}$. Denote $L_i = \frac{\lambda_i}{\lambda_i + \kappa_{\mathsf{TE}}}$ for $i \in [d]$. Taking expectation over the randomness of $\mathbf{U}_{\mathsf{TE}}$, we have*

$$\mathbb{E}[\mathbf{P}_{\mathsf{TE}}] \simeq \mathrm{diag}\left\{ \frac{\lambda_1}{\lambda_1 + \kappa_{\mathsf{TE}}}, \ldots, \frac{\lambda_d}{\lambda_d + \kappa_{\mathsf{TE}}} \right\}, \tag{14}$$

*and*

$$\mathrm{Cov}[(\mathbf{P}_{\mathsf{TE}})_{ij}, (\mathbf{P}_{\mathsf{TE}})_{k\ell}] \simeq \frac{L_i(1 - L_j)L_k(1 - L_\ell)}{M_{\mathsf{TE}} - \sum_{i' \in [d]} L_{i'}^2}(\mathbf{1}_{ik}\mathbf{1}_{j\ell} + \mathbf{1}_{i\ell}\mathbf{1}_{jk} - \mathbf{1}_{ij}\mathbf{1}_{k\ell}), \quad i, j, k, \ell \in [d], \tag{15}$$

*where $\mathbf{1}_{ij} = \mathbf{1}\{i = j\}$ for $i, j \in [d]$.*

*Proof of Lemma A.1.* Define $L_i = \frac{\lambda_i}{\lambda_i + \kappa_{\mathsf{TE}}}$, next we will derive the covariance matrix. Differentiating both sides of $\sum_{i=1}^{d} \frac{\lambda_i}{\lambda_i + \kappa_{\mathsf{TE}}} = M_{\mathsf{TE}}$ with respect to $\lambda_i$, we find

$$\frac{\partial}{\partial \lambda_i} \sum_{j=1}^{d} \frac{\lambda_j}{\lambda_j + \kappa_{\mathsf{TE}}} = \sum_{j=1}^{d} \left( -\frac{\lambda_j}{(\lambda_j + \kappa)^2} \frac{\partial \kappa_{\mathsf{TE}}}{\partial \lambda_i} \right) + \frac{\kappa_{\mathsf{TE}}}{(\lambda_i + \kappa_{\mathsf{TE}})^2} = 0,$$

which yields that

$$\frac{\partial \kappa_{\mathsf{TE}}}{\partial \lambda_i} = \frac{\kappa_{\mathsf{TE}}^2}{q(\lambda_i + \kappa_{\mathsf{TE}})^2}, \quad \text{where} \quad q = \sum_{j=1}^{d} \frac{\kappa_{\mathsf{TE}} \lambda_j}{(\lambda_j + \kappa_{\mathsf{TE}})^2}. \tag{16}$$

Recall that

$$\mathbf{P}_{\mathsf{TE}} = \Lambda^{1/2} \mathbf{Z} \Lambda^{1/2} \quad \text{where} \quad \mathbf{Z} = \mathbf{U}_{\mathsf{TE}} \left( \mathbf{U}_{\mathsf{TE}}^\top \Lambda \mathbf{U}_{\mathsf{TE}} \right)^{-1} \mathbf{U}_{\mathsf{TE}}^\top \in \mathbb{R}^{d \times d}. \tag{17}$$

Due to the assumption $M_{\mathsf{TE}}/d < 1$, we can apply Theorem 4.1 in LeJeune et al. (2024) (Eq. (4.3) and (4.4) with $p = d$ and $q = M_{\mathsf{TE}}$) such that

$$\mathbf{Z} \simeq (\Lambda + \kappa_{\mathsf{TE}} \mathbf{I}_d)^{-1} = \mathrm{diag}\left\{ \frac{1}{\lambda_1 + \kappa_{\mathsf{TE}}}, \ldots, \frac{1}{\lambda_d + \kappa_{\mathsf{TE}}} \right\}. \tag{18}$$

Hence conclusion (14) follows immediately by recalling $\Lambda = \mathrm{diag}\{\lambda_1, \ldots, \lambda_d\}$. Let $\varphi_i$ be the $i$-th row of $\mathbf{U}_{\mathsf{TE}}$ for $i \in [d]$. Differentiating $\mathbf{Z} = \mathbf{U}_{\mathsf{TE}} \left( \mathbf{U}_{\mathsf{TE}}^\top \Lambda \mathbf{U}_{\mathsf{TE}} \right)^{-1} \mathbf{U}_{\mathsf{TE}}^\top$ w.r.t. $\Lambda_{jk}$ gives: for $i, \ell \in [d]$,

$$\frac{\partial \mathbf{Z}_{i\ell}}{\partial \Lambda_{jk}} = -\varphi_i^\top \left( \mathbf{U}_{\mathsf{TE}}^\top \Lambda \mathbf{U}_{\mathsf{TE}} \right)^{-1} \varphi_j \varphi_k^\top \left( \mathbf{U}_{\mathsf{TE}}^\top \Lambda \mathbf{U}_{\mathsf{TE}} \right)^{-1} \varphi_\ell = -\mathbf{Z}_{ij} \mathbf{Z}_{k\ell}.$$

So we have for $i, j, k, \ell \in [d]$,

$$\mathbb{E}[\mathbf{Z}_{ij} \mathbf{Z}_{k\ell}] = -\frac{\partial}{\partial \Lambda_{jk}} \mathbb{E}[\mathbf{Z}_{i\ell}]. \tag{19}$$

Setting $\ell = i, k = j$ in (19), and using the fact $\Lambda_{jj} = \lambda_j$ and (18), we obtain:

$$\mathbb{E}[\mathbf{Z}_{ij}^2] = \mathbb{E}[\mathbf{Z}_{ij} \mathbf{Z}_{ji}] = -\frac{\partial}{\partial \lambda_j} \mathbb{E}[\mathbf{Z}_{ii}] \simeq -\frac{\partial}{\partial \lambda_j} \left( \frac{1}{\lambda_i + \kappa_{\mathsf{TE}}} \right)$$

$$= \frac{1}{(\lambda_i + \kappa_{\mathsf{TE}})^2} \left( \mathbf{1}_{ij} + \frac{\partial \kappa_{\mathsf{TE}}}{\partial \lambda_j} \right), \tag{20}$$

where we used the fact $\mathbf{Z}_{ij} = \mathbf{Z}_{ji}$ since $Z$ is symmetric. Together with (16), for $i \neq j$ we have

$$\mathrm{Cov}[\mathbf{Z}_{ij}, \mathbf{Z}_{ji}] = \mathrm{Cov}[\mathbf{Z}_{ij}, \mathbf{Z}_{ij}] = \mathbb{E}[\mathbf{Z}_{ij}^2] - \mathbb{E}[\mathbf{Z}_{ij}]^2$$

$$\simeq \frac{1}{(\lambda_i + \kappa_{\mathsf{TE}})^2} \cdot \frac{\partial \kappa_{\mathsf{TE}}}{\partial \lambda_j}$$

$$= \frac{\kappa_{\mathsf{TE}}^2}{q(\lambda_i + \kappa_{\mathsf{TE}})^2(\lambda_j + \kappa_{\mathsf{TE}})^2}, \tag{21}$$

where we used $\mathbb{E}[\mathbf{Z}_{ij}] = 0$ if $i \neq j$.

Given a deterministic orthogonal matrix $\mathbf{U} \in \mathbb{R}^{d \times d}$, we define

$$\mathbf{Z}^{(\mathbf{U})} = \mathbf{U}_{\mathsf{TE}} \left( \mathbf{U}_{\mathsf{TE}}^{\top} U^{\top} \Lambda \mathbf{U} \mathbf{U}_{\mathsf{TE}} \right)^{-1} \mathbf{U}_{\mathsf{TE}}^{\top}.$$

Due to (18), we have

$$\mathbb{E}[\mathbf{Z}^{(\mathbf{U})}] \simeq \mathbf{U}^{\top} (\Lambda + \kappa_{\mathsf{TE}} \mathbf{I})^{-1} \mathbf{U}.$$

For $i \neq j$, we consider:

$$\left( \frac{\partial}{\partial \mathbf{U}_{ij}} - \frac{\partial}{\partial \mathbf{U}_{ji}} \right) \mathbf{Z}_{ij}^{(U)} \bigg|_{\mathbf{U}=\mathbf{I}}$$

$$= -\varphi_i^{\top} \left( \mathbf{U}_{\mathsf{TE}}^{\top} \Lambda \mathbf{U}_{\mathsf{TE}} \right)^{-1} \left( \varphi_j \lambda_i \varphi_i^{\top} - \varphi_i \lambda_j \varphi_j^{\top} + \varphi_i \lambda_i \varphi_j^{\top} - \varphi_j \lambda_j \varphi_i^{\top} \right) \left( \mathbf{U}_{\mathsf{TE}}^{\top} \Lambda \mathbf{U}_{\mathsf{TE}} \right)^{-1} \varphi_j$$

$$= (\lambda_j - \lambda_i)(\mathbf{Z}_{ij}^2 + \mathbf{Z}_{ii} \mathbf{Z}_{jj}). \tag{22}$$

In addition, notice that

$$\left( \frac{\partial}{\partial \mathbf{U}_{ij}} - \frac{\partial}{\partial \mathbf{U}_{ji}} \right) \mathbb{E}[\mathbf{Z}_{ij}^{(\mathbf{U})}] \bigg|_{\mathbf{U}=\mathbf{I}} \simeq \left( \frac{\partial}{\partial \mathbf{U}_{ij}} - \frac{\partial}{\partial \mathbf{U}_{ji}} \right) [\mathbf{U}^{\top} (\Lambda + \kappa_{\mathsf{TE}} \mathbf{I})^{-1} \mathbf{U}]_{ij} \bigg|_{\mathbf{U}=\mathbf{I}}$$

$$\simeq \frac{1}{\lambda_i + \kappa_{\mathsf{TE}}} - \frac{1}{\lambda_j + \kappa_{\mathsf{TE}}}.$$

Taking the expectation of (22) leads to

$$(\lambda_j - \lambda_i))(\mathbb{E}[\mathbf{Z}_{ii} \mathbf{Z}_{jj}] + \mathbb{E}[\mathbf{Z}_{ij}^2]) \simeq \frac{1}{\lambda_i + \kappa_{\mathsf{TE}}} - \frac{1}{\lambda_j + \kappa_{\mathsf{TE}}}.$$

Plugging the relation (20) for $\mathbb{E}[\mathbf{Z}_{ij}^2]$ into the equation above, we get

$$\mathbb{E}[\mathbf{Z}_{ii} \mathbf{Z}_{jj}] \simeq \frac{1}{(\lambda_i + \kappa_{\mathsf{TE}})(\lambda_j + \kappa_{\mathsf{TE}})} - \mathbb{E}[\mathbf{Z}_{ij}^2]$$

$$\simeq \frac{1}{(\lambda_i + \kappa_{\mathsf{TE}})(\lambda_j + \kappa_{\mathsf{TE}})} - \frac{\kappa_{\mathsf{TE}}^2}{q(\lambda_i + \kappa_{\mathsf{TE}})^2 (\lambda_j + \kappa_{\mathsf{TE}})^2}.$$

Then we can compute the covariance as

$$\mathrm{Cov}[\mathbf{Z}_{ii}, \mathbf{Z}_{jj}] = \mathbb{E}[\mathbf{Z}_{ii} \mathbf{Z}_{jj}] - \mathbb{E}[\mathbf{Z}_{ii}] \mathbb{E}[\mathbf{Z}_{jj}] = -\frac{\kappa_{\mathsf{TE}}^2}{q(\lambda_i + \kappa_{\mathsf{TE}})^2 (\lambda_j + \kappa_{\mathsf{TE}})^2} \tag{23}$$

**Summary.** By observing (21) and (23), the covariance matrix can be summarized as:

$$\mathrm{Cov}[\mathbf{Z}_{ij}, \mathbf{Z}_{k\ell}] \simeq \frac{\kappa_{\mathsf{TE}}^2}{q} \cdot \frac{\mathbf{1}_{ik} \mathbf{1}_{j\ell} + \mathbf{1}_{i\ell} \mathbf{1}_{jk} - \mathbf{1}_{ij} \mathbf{1}_{k\ell}}{(\lambda_i + \kappa_{\mathsf{TE}})(\lambda_j + \kappa_{\mathsf{TE}})(\lambda_k + \kappa_{\mathsf{TE}})(\lambda_\ell + \kappa_{\mathsf{TE}})}. \tag{24}$$

Recall that $\mathbf{P}_{ij} = \lambda_i \mathbf{Z}_{ij}$, and $L_i = \frac{\lambda_i}{\lambda_i + \kappa_{\mathsf{TE}}}$, $q = \sum_{i \in [d]} L_i (1 - L_i)$, we find

$$\mathrm{Cov}[\mathbf{P}_{ij}, \mathbf{P}_{k\ell}] \simeq \frac{L_i (1 - L_j) L_k (1 - L_\ell)}{M_{\mathsf{TE}} - \sum_{i \in [d]} L_i^2} (\mathbf{1}_{ik} \mathbf{1}_{j\ell} + \mathbf{1}_{i\ell} \mathbf{1}_{jk} - \mathbf{1}_{ij} \mathbf{1}_{k\ell}),$$

where we also used the fact $\sum_{i \in [d]} L_i = M_{\mathsf{TE}}$. Hence we have proved the relation (15). $\square$

## A.2 STUDENT MODEL

### A.2.1 PROOF OF LEMMA 4.8

*Proof.* Using the definition $y_{\mathsf{TE}} = x^{\top} \mathbf{U}_{\mathsf{TE}} w_{\mathsf{TE}} = x^{\top} \mathbf{U}_{\mathsf{TE}} \left( \mathbf{U}_{\mathsf{TE}}^{\top} \Lambda \mathbf{U}_{\mathsf{TE}} \right)^{-1} \mathbf{U}_{\mathsf{TE}}^{\top} \Lambda \theta^*$, we have

$$\mathbb{E}_{x \sim \mathcal{D}} \left[ \left( y_{\mathsf{TE}} - x^{\top} \mathbf{U}_{\mathsf{ST}} w \right) \right]$$

$$= \mathbb{E}_{x \sim \mathcal{D}} \left[ \left( x^{\top} \mathbf{U}_{\mathsf{TE}} \left( \mathbf{U}_{\mathsf{TE}}^{\top} \Lambda \mathbf{U}_{\mathsf{TE}} \right)^{-1} \mathbf{U}_{\mathsf{TE}}^{\top} \Lambda \theta^* - x^{\top} \mathbf{U}_{\mathsf{ST}} w \right)^2 \right]$$

$$= (\theta^*)^{\top} \Lambda \mathbf{U}_{\mathsf{ST}} \left( \mathbf{U}_{\mathsf{TE}}^{\top} \Lambda \mathbf{U}_{\mathsf{TE}} \right)^{-1} \mathbf{U}_{\mathsf{ST}}^{\top} \Lambda \theta^* - 2 w^{\top} \mathbf{U}_{\mathsf{ST}} \Lambda \mathbf{U}_{\mathsf{TE}} \left( \mathbf{U}_{\mathsf{TE}}^{\top} \Lambda \mathbf{U}_{\mathsf{TE}} \right)^{-1} \mathbf{U}_{\mathsf{TE}}^{\top} \Lambda \theta^* - w^{\top} \mathbf{U}_{\mathsf{ST}}^{\top} \Lambda \mathbf{U}_{\mathsf{ST}} w$$

$$= \|\mathbf{P}_{\mathsf{TE}} v^*\|^2 - 2 (\Lambda^{1/2} \mathbf{U}_{\mathsf{TE}} w)^{\top} \mathbf{P}_{\mathsf{TE}} v^* + \|\Lambda^{1/2} \mathbf{U}_{\mathsf{TE}} w\|^2$$

$$= \|v_{\mathsf{TE}} - \Lambda^{1/2} \mathbf{U}_{\mathsf{TE}} w\|^2. \tag{25}$$

$\square$

### A.2.2 PROOF OF LEMMA 4.9

*Proof.* By Lemma 4.8, we can write the gradient flow as

$$\frac{d\boldsymbol{w}_{\mathsf{ST}}(t)}{dt} = -\frac{d}{d\boldsymbol{w}} \mathbb{E}_{x \sim \mathcal{D}} \left[ \left( y_{\mathsf{TE}} - x^\top \mathbf{U}_{\mathsf{ST}} \boldsymbol{w} \right) \right] \Big|_{\boldsymbol{w} = \boldsymbol{w}_{\mathsf{ST}}(t)}$$

$$= -\frac{d}{d\boldsymbol{w}} \left\| v_{\mathsf{TE}} - \Lambda^{1/2} \mathbf{U}_{\mathsf{ST}} \boldsymbol{w} \right\|^2 \Big|_{\boldsymbol{w} = \boldsymbol{w}_{\mathsf{ST}}(t)}$$

$$= 2 U_{\mathsf{ST}} \Lambda^{1/2} \left( v_{\mathsf{TE}} - \Lambda^{1/2} \mathbf{U}_{\mathsf{ST}} \boldsymbol{w}_{\mathsf{ST}}(t) \right). \tag{26}$$

From Lemma 4.8, student model is given by ($\mathbf{U}_{\mathsf{ST}}$ is fixed gaussian projection)

$$\boldsymbol{w}_{\mathsf{ST}} = \arg\min_{\boldsymbol{w} \in \mathbb{R}^{M_{\mathsf{ST}}}} \left\| v_{\mathsf{TE}} - \Lambda^{1/2} \mathbf{U}_{\mathsf{ST}} \boldsymbol{w} \right\|^2. \tag{27}$$

From Eq. (26), we know that gradient flow dynamics is

$$\frac{d\boldsymbol{w}_{\mathsf{ST}}(t)}{dt} = -2\mathbf{U}_{\mathsf{ST}}^\top \Lambda^{1/2} \left( \Lambda^{1/2} \mathbf{U}_{\mathsf{ST}} \boldsymbol{w}_{\mathsf{ST}}(t) - v_{\mathsf{TE}} \right) = -2\mathbf{U}_{\mathsf{ST}}^\top \Lambda^{1/2} \left( v_{\mathsf{ST}}(t) - v_{\mathsf{TE}} \right).$$

According to Bai-Yin's law stated in Lemma C.1, we can guarantee

$$\lambda_{\max}(\mathbf{U}_{\mathsf{ST}} \mathbf{U}_{\mathsf{ST}}^\top - \mathbf{I}_d) = \lambda_{\max}(\mathbf{U}_{\mathsf{ST}})^2 - 1 = \frac{1}{M_{\mathsf{ST}}} \left( \sqrt{M_{\mathsf{ST}}} + \sqrt{d} + o(\sqrt{d}) \right)^2$$

$$= 1 + O\left( \frac{d}{M_{\mathsf{ST}}} \right),$$

which means $\lim_{(d, M_{\mathsf{ST}}) \to \infty} \lambda_{\max}(\mathbf{U}_{\mathsf{ST}} \mathbf{U}_{\mathsf{ST}}^\top - \mathbf{I}_d) = 0$ almost surely under the assumption $d/M_{\mathsf{ST}} \to 0$ and the Gaussian random projection. Then we have

$$\frac{d}{dt} v_{\mathsf{ST}}(t) = -2\Lambda^{1/2} \mathbf{U}_{\mathsf{ST}} \mathbf{U}_{\mathsf{ST}}^\top \Lambda^{1/2} \left( v_{\mathsf{ST}}(t) - v_{\mathsf{TE}} \right) \tag{28}$$

$$\overset{(i)}{\simeq} -2 \left( \sum_{i=1}^d \lambda_i \boldsymbol{e}_i \boldsymbol{e}_i^\top \right) \left( v_{\mathsf{ST}}(t) - v_{\mathsf{TE}} \right)$$

$$= -2 \sum_{i=1}^d \lambda_i \langle v_{\mathsf{ST}}(t) - v_{\mathsf{TE}}, \boldsymbol{e}_i \rangle \cdot \boldsymbol{e}_i,$$

where (i) due to $\mathbf{U}_{\mathsf{ST}} \mathbf{U}_{\mathsf{ST}}^\top \simeq \mathbf{I}_d$. Solving the following ODE with initializing at $v_{\mathsf{ST}}(0) = 0$,

$$\frac{d}{dt} \langle v_{\mathsf{ST}}(t) - v_{\mathsf{TE}}, \boldsymbol{e}_i \rangle = -2\lambda_i \langle v_{\mathsf{ST}}(t) - v_{\mathsf{TE}}, \boldsymbol{e}_i \rangle,$$

we can get the solution

$$\langle v_{\mathsf{ST}}(t), \boldsymbol{e}_i \rangle - \langle v_{\mathsf{TE}}, \boldsymbol{e}_i \rangle = e^{-2\lambda_i t} (\langle v_{\mathsf{ST}}(0), \boldsymbol{e}_i \rangle - \langle v_{\mathsf{TE}}, \boldsymbol{e}_i \rangle).$$

The ODE solution for the $i$-th coordinate is given by:

$$[v_{\mathsf{ST}}(t)]_i \simeq (1 - e^{-2\lambda_i t})(v_{\mathsf{TE}})_i, \quad i \in [d].$$

It means that

$$v_{\mathsf{ST}}(t) \simeq \text{diag} \left\{ 1 - e^{-2\lambda_1 t}, \ldots, 1 - e^{-2\lambda_d t} \right\} v_{\mathsf{TE}} = \boldsymbol{\gamma}(t) \odot v_{\mathsf{TE}},$$

where $\boldsymbol{\gamma}(t) := (\gamma_1(t), \ldots, \gamma_d(t))$ with $\gamma_i(t) := 1 - e^{-2\lambda_i t}$. $\qquad \square$

By comparing Theorems 4.1 and 4.2, for each coordinate $i \in [d]$, if

$$(L_i - \mathcal{L}_i(t))(2 - L_i - \mathcal{L}_i(t))(v_i^*)^2 \leq \frac{L_i^2 - \mathcal{L}_i(t)^2}{M_{\mathsf{TE}} - \sum_{j \in [d]} L_j^2} \sum_{j \in [d]} (1 - L_j)^2 (v_j^*)^2,$$

i.e.,

$$(v_i^*)^2 \leq \frac{L_i + \mathcal{L}_i(t)}{2 - L_i - \mathcal{L}_i(t)} \cdot \frac{\sum_{j \in [d]} (1 - L_j)^2 (v_j^*)^2}{M_{\mathsf{TE}} - \sum_{j \in [d]} L_j^2},$$

then we have $\mathcal{E}_{\mathsf{ST}}(t)/\mathcal{E}_{\mathsf{TE}} \leq 1$.

### A.2.3 PROOF OF THEOREM 4.2

*Proof of Theorem 4.2.* Let us denote $\mathcal{L}_i(t) = \gamma_i(t)L_i$, then the MSE of student model at time $t$ is given by

$$
\begin{aligned}
\mathcal{E}_{\mathsf{st}}(t) &= \mathbb{E}\left[\left(x^\top \boldsymbol{\theta}^* - x^\top \mathbf{U}_{\mathsf{ST}}\boldsymbol{w}_{\mathsf{ST}}(t)\right)^2\right] \\
&= \mathbb{E}\left[(\boldsymbol{\theta}^*)^\top \Lambda \boldsymbol{\theta}^* - 2(\boldsymbol{\theta}^*)^\top \Lambda \mathbf{U}_{\mathsf{ST}}\boldsymbol{w}_{\mathsf{ST}}(t) + (\mathbf{U}_{\mathsf{ST}}\boldsymbol{w}_{\mathsf{ST}}(t))^\top \Lambda \mathbf{U}_{\mathsf{ST}}\boldsymbol{w}_{\mathsf{ST}}(t)\right] \\
&= \mathbb{E}\left[\|v^* - v_{\mathsf{ST}}(t)\|^2\right] \\
&= \|v^*\|^2 - 2(v^*)^\top \mathbb{E}\left[v_{\mathsf{ST}}(t)\right] + \mathbb{E}\left[\|v_{\mathsf{ST}}(t)\|^2\right] \\
&= \|v^*\|^2 - 2(v^*)^\top \mathbb{E}\left[\boldsymbol{\gamma}(t) \odot v_{\mathsf{TE}}(t)\right] + \mathbb{E}\left[\|\boldsymbol{\gamma}(t) \odot v_{\mathsf{TE}}(t)\|^2\right] \\
&\overset{(i)}{\simeq} \sum_{i\in[d]} (v_i^*)^2 - 2v_i^* \gamma_i(t)\mathbb{E}[(v_{\mathsf{TE}})_i] + \gamma_i^2(t)\mathbb{E}\left[(v_{\mathsf{TE}})_i\right]^2 + \gamma_i^2(t)\operatorname{Var}[(v_{\mathsf{TE}})_i] \\
&\overset{(ii)}{\simeq} \sum_{i\in[d]} (v_i^*)^2 - 2v_i^* \gamma_i(t)L_i v_i^* + \gamma_i^2(t)(L_i v_i^*)^2 + \frac{\gamma_i^2(t)L_i^2}{M_{\mathsf{TE}} - \sum_{j\in[d]} L_j^2} \sum_{j\in[d]} (1 - L_j)^2 (v_j^*)^2 \\
&= \sum_{i\in[d]} (1 - \mathcal{L}_i(t))^2 (v_i^*)^2 + \frac{\mathcal{L}_i^2(t)}{M_{\mathsf{TE}} - \sum_{j\in[d]} L_j^2} \sum_{j\in[d]} (1 - L_j)^2 (v_j^*)^2 \\
&\overset{(iii)}{=} \sum_{i\in[d]} (1 - \mathcal{L}_i(t))^2 (v_i^*)^2 + \frac{\mathcal{L}_i^2(t)}{M_{\mathsf{TE}}} \mathcal{E}_{\mathsf{TE}} \\
&= \sum_{i\in[d]} (1 - \gamma_i(t)L_i)^2 (v_i^*)^2 + \frac{\gamma_i(t)^2 L_i^2}{M_{\mathsf{TE}}} \mathcal{E}_{\mathsf{TE}} \\
&= \sum_{i\in[d]} \left(\frac{L_i^2 \mathcal{E}_{\mathsf{TE}}}{M_{\mathsf{TE}}} + L_i^2 (v_i^*)^2\right) \gamma_i^2(t) - 2L_i (v_i^*)^2 \gamma_i(t) + (v_i^*)^2 \\
&:= \sum_{i\in[d]} A_i \left(\gamma_i(t) - \frac{B_i}{A_i}\right)^2 + (v_i^*)^2 - \frac{B_i^2}{A_i}, \quad (29)
\end{aligned}
$$

where (i) holds due to (12) and (13); (ii) follows from Theorem 4.1; (iii) holds due to (5); and

$$
A_i = \frac{L_i^2 \mathcal{E}_{\mathsf{TE}}}{M_{\mathsf{TE}}} + L_i^2 (v_i^*)^2 \quad B_i = L_i (v_i^*)^2.
$$

$\square$

### A.3 PROOF OF THEOREM 4.3

Theorem 4.3 is a special case of the following theorem with $r = 3/2$ and $p = 1/2$.

**Theorem A.2.** *Let $M_{\mathsf{TE}} = m$, $d = m^r$ and $K = m^p$ for $r \in (1, 2)$ and $p \in [0, 1)$, and $d/M_{\mathsf{ST}} \to 0$ when $m \to \infty$, where $K$ is the number of large eigenvalues. In particular, we set the covariance matrix as $\lambda_i = 1$ if $i \le K$ and $\lambda_i = m^{-1}$ for $K < i \le d$, and choose the ground truth model $\boldsymbol{\theta}^*$ such that $(v_i^*)^2 = \lambda_i(\theta_i^*)^2 = m^\alpha$ if $i \le K$ and $(v_i^*)^2 = \lambda_i(\theta_i^*)^2 = m^{-\beta}$ for $K < i \le d$, where $\alpha, \beta \ge 0$. Suppose the parameters satisfy $r - p \ge 1$, $\alpha + p + r \ge 3$, and $4 < r + p + \alpha + \beta \le 5$, then there exists time $T$ such that*

$$
\frac{\mathcal{E}_{\mathsf{st}}(T)}{\mathcal{E}_{\mathsf{TE}}} \le \Theta(m^{-1+p}) + \Theta(m^{-(r+p+\alpha+\beta)+4}).
$$

*Proof.* Applying the upper bound in Lemma C.2 with $k = 1$, we can have

$$
\begin{aligned}
\kappa_{\mathsf{TE}} &\le \frac{\sum_{i=1}^d \lambda_i}{M_{\mathsf{TE}} - 1} = \frac{K + (d - K)m^{-1}}{M_{\mathsf{TE}} - 1} = \frac{m^p + (m^r - m^p)m^{-1}}{M_{\mathsf{TE}} - 1} \\
&\le \frac{m^p + m^{r-1}}{M_{\mathsf{TE}} - 1} = \Theta\left(m^{p-1} + m^{r-2}\right) = \Theta(m^{r-2})(1 + \Theta(m^{p-r+1})).
\end{aligned}
$$

Applying the lower bound in Lemma C.2 with $k = 1$, we can have

$$\kappa_{\mathsf{TE}} \geq \lambda_d \left( \frac{d}{M_{\mathsf{TE}}} - 1 \right) = \frac{1}{m} \left( \frac{m^r}{M_{\mathsf{TE}}} - 1 \right) = \Theta(m^{r-2}).$$

The two bounds above imply that $\kappa_{\mathsf{TE}} = \Theta(m^{r-2}) = o(1)$ due to $r - p \geq 1$ and $r \in (1, 2)$. Recalling the definition of $L_i$, we know

$$L_i = \frac{\lambda_i}{\lambda_i + \kappa_{\mathsf{TE}}} = \begin{cases} 1/(1 + \kappa_{\mathsf{TE}}) & 1 \leq i \leq K \\ 1/(1 + m\kappa_{\mathsf{TE}}) & K < i \leq d \end{cases},$$

which implies

$$\sum_{i \in [d]} L_i^2 = \sum_{i=1}^K \frac{1}{(1 + \kappa_{\mathsf{TE}})^2} + \sum_{i=K+1}^d \frac{1}{(1 + m\kappa_{\mathsf{TE}})^2} = \frac{m^p}{(1 + \kappa_{\mathsf{TE}})^2} + \frac{m^r - m^p}{(1 + m\kappa_{\mathsf{TE}})^2} = \Theta(m^p) + \Theta(m^{2-r}).$$

Recalling the MSE of the teacher model in Theorem 4.1, we have

$$\mathcal{E}_{\mathsf{TE}} = \frac{1}{1 - \frac{1}{M_{\mathsf{TE}}} \sum_i L_i^2} \left( \sum_{i \in [K]} (1 - L_i)^2 (v_i^*)^2 + \sum_{i > K} (1 - L_i)^2 (v_i^*)^2 \right)$$

$$= \frac{1}{1 - \Theta(m^{p-1}) - \Theta(m^{1-r})} \left( \frac{m^p \kappa_{\mathsf{TE}}^2 m^\alpha}{(1 + \kappa_{\mathsf{TE}})^2} + \frac{m^{2-\beta}(m^r - m^p)\kappa_{\mathsf{TE}}^2}{(1 + m\kappa_{\mathsf{TE}})^2} \right)$$

$$\overset{(i)}{=} \Theta(m^{2r-4+p+\alpha}) + \Theta(m^{r-\beta}) = \Theta(m^{2r-4+p+\alpha}) \overset{(ii)}{\geq} \Theta(m^{r-1}). \tag{30}$$

where (i) follows from $\kappa_{\mathsf{TE}} = \Theta(m^{r-2}) = o(1)$, and $p \in (0, 1), r \in (1, 2)$; and (ii) holds due to $\alpha + p + r \geq 3$ and $r > 1$. We choose the time $T$ such that for $i \leq K$,

$$\gamma_i(T) = 1 - e^{-2T} = \frac{1 + \kappa_{\mathsf{TE}}}{\frac{\mathcal{E}_{\mathsf{TE}}}{M_{\mathsf{TE}}} + 1} \implies T = -\frac{1}{2} \log \left( \frac{\mathcal{E}_{\mathsf{TE}} - \kappa_{\mathsf{TE}} m}{\mathcal{E}_{\mathsf{TE}} + m} \right) = O(\log m).$$

At time $T$, we have for $K < i \leq d$,

$$\gamma_i(T) = 1 - e^{-2T/m} \leq \frac{1}{m} \log \left( \frac{\mathcal{E}_{\mathsf{TE}} - \kappa_{\mathsf{TE}} m}{\mathcal{E}_{\mathsf{TE}} + m} \right),$$

where we used inequality $1 - e^{-x} \leq x$. It follows that for $i > K$,

$$\frac{B_i}{A_i} = \frac{L_i (v_i^*)^2}{\frac{L_i^2 \mathcal{E}_{\mathsf{TE}}}{M_{\mathsf{TE}}} + L_i^2 (v_i^*)^2} = \frac{m^{-\beta}(1 + m\kappa_{\mathsf{TE}})}{\frac{\mathcal{E}_{\mathsf{TE}}}{m} + m^{-\beta}} = \frac{1 + m\kappa_{\mathsf{TE}}}{1 + m^{\beta-1}\mathcal{E}_{\mathsf{TE}}}$$

$$\overset{(i)}{=} \Theta \left( \frac{1}{m^{\beta+r+p+\alpha-4}} \right) \overset{(ii)}{>} \frac{1}{m} \log \left( \frac{\mathcal{E}_{\mathsf{TE}} - \kappa_{\mathsf{TE}} m}{\mathcal{E}_{\mathsf{TE}} + m} \right) \geq \gamma_i(T),$$

where (i) holds since $\mathcal{E}_{\mathsf{TE}} = \Theta(m^{2r+p-4+\alpha})$, see (30), and $\kappa_{\mathsf{TE}} = \Theta(m^{r-2})$; (ii) holds due to $\beta + \alpha + p + r < 5$. Then at time $T$, we have

$$\sum_{i>K}^d A_i \left( \gamma_i(T) - \frac{B_i}{A_i} \right)^2 + (v_i^*)^2 - \frac{B_i^2}{A_i} \leq \sum_{i>K}^d A_i \left( \gamma_i(0) - \frac{B_i}{A_i} \right)^2 + (v_i^*)^2 - \frac{B_i^2}{A_i} \leq \sum_{i>K}^d (v_i^*)^2, \tag{31}$$

where we used $\gamma_i(0) = 0$ and $\gamma_i(t)$ is increasing over $t$. In addition, we notice that for any $i \leq K$,

$$\frac{B_i^2}{A_i} = \frac{L_i^2 (v_i^*)^4}{\frac{L_i^2 \mathcal{E}_{\mathsf{TE}}}{M_{\mathsf{TE}}} + L_i^2 (v_i^*)^2} = \frac{m^{2\alpha}}{\mathcal{E}_{\mathsf{TE}}/m + m^\alpha} = \frac{m^\alpha}{m^{2r+p-5} + 1} \tag{32}$$

where we use $\mathcal{E}_{\mathsf{TE}} = \Theta(m^{2r+p+\alpha-4})$, and

$$v_i^{*2} - \frac{B_i^2}{A_i} = m^\alpha - \frac{m^\alpha}{m^{2r+p-5} + 1} = \frac{m^{2r+p+\alpha-5}}{m^{2r+p-5} + 1} = \Theta(m^{2r+p+\alpha-5}) \tag{33}$$

which holds due to $\alpha > 0$ and $2r + p < 5$. From Eq. 33, the MSE of the student model at time $T$ satisfies

$$
\begin{aligned}
\mathcal{E}_{\mathsf{st}}(T) &= \sum_{i \in [d]} A_i \left( \gamma_i(T) - \frac{B_i}{A_i} \right)^2 + (v_i^*)^2 - \frac{B_i^2}{A_i} \\
&= \sum_{i=1}^{K} (v_i^*)^2 - \frac{B_i^2}{A_i} + \sum_{i > K} A_i \left( \gamma_i(T) - \frac{B_i}{A_i} \right)^2 + (v_i^*)^2 - \frac{B_i^2}{A_i} \\
&\overset{(i)}{\leq} \sum_{i=1}^{K} (v_i^*)^2 - \frac{B_i^2}{A_i} + \sum_{i > K}^{d} (v_i^*)^2 \\
&\overset{(ii)}{=} \Theta(m^{2r+2p+\alpha-5}) + \Theta \left( \frac{m^r - m^p}{m^\beta} \right) \\
&= \Theta(m^{2r+2p+\alpha-5}) + \Theta(m^{r-\beta}),
\end{aligned}
\tag{34}
$$

where (i) follows from (31). (ii) follows from (33) and $K = m^p$, Using $\mathcal{E}_{\mathsf{TE}} = \Theta(m^{2r+p+\alpha-4})$ again, we can compute bound the ratio as

$$
\frac{\mathcal{E}_{\mathsf{st}}(T)}{\mathcal{E}_{\mathsf{TE}}} \leq \Theta(m^{-1+p}) + \Theta(m^{-(r+p+\alpha+\beta)+4}).
\tag{35}
$$

$\square$

### A.4 PROOF OF THEOREM 4.4

*Proof.* By Theorem 4.2, the MSE of the student model satisfies that for any $T \geq 0$

$$
\begin{aligned}
\mathcal{E}_{\mathsf{st}}(T) &\simeq \sum_{i \in [d]} A_i \left( \gamma_i(t) - \frac{B_i}{A_i} \right)^2 + (v_i^*)^2 - \frac{B_i^2}{A_i} \\
&\geq \sum_{i \in [d]} (v_i^*)^2 - \frac{B_i^2}{A_i} \\
&= \sum_{i \in [d]} (v_i^*)^2 - \frac{L_i^2 (v_i^*)^4}{\frac{L_i^2 \mathcal{E}_{\mathsf{TE}}}{M_{\mathsf{TE}}} + L_i^2 (v_i^*)^2} \\
&= \sum_{i \in [d]} (v_i^*)^2 \left( 1 - \frac{M_{\mathsf{TE}} (v_i^*)^2}{\mathcal{E}_{\mathsf{TE}} + M_{\mathsf{TE}} (v_i^*)^2} \right) \\
&= \sum_{i \in [d]} \frac{\mathcal{E}_{\mathsf{TE}}}{\mathcal{E}_{\mathsf{TE}}/(v_i^*)^2 + M_{\mathsf{TE}}},
\end{aligned}
$$

which gives the teacher-student-MSE ratio

$$
\frac{\mathcal{E}_{\mathsf{st}}(T)}{\mathcal{E}_{\mathsf{TE}}} \geq \sum_{i \in [d]} \frac{1}{\mathcal{E}_{\mathsf{TE}}/(v_i^*)^2 + M_{\mathsf{TE}}}.
\tag{36}
$$

Applying Lemma C.2, we know

$$
\kappa_{\mathsf{TE}} \geq \lambda_k \left( \frac{k}{M_{\mathsf{TE}}} - 1 \right), \quad \forall k \geq M_{\mathsf{TE}}.
$$

Together with Theorem 4.1, we have

$$
\begin{aligned}
\mathcal{E}_{\mathsf{TE}} &\simeq \frac{M_{\mathsf{TE}}}{M_{\mathsf{TE}} - \sum_{i=1}^{d} L_i^2} \sum_{i=1}^{d} (1 - L_i)^2 (v_i^*)^2 \\
&\geq \sum_{i=1}^{d} \left( 1 - \frac{\lambda_i}{\lambda_i + \kappa_{\mathsf{TE}}} \right)^2 (v_i^*)^2
\end{aligned}
$$

$$\geq \sum_{i=M_{\mathsf{TE}}}^{d} \left(1 - \frac{M_{\mathsf{TE}}}{i}\right)^2 (v_i^*)^2.$$

$\square$

# B  DEFERRED PROOFS OF RELU NEURAL NETWORKS

## B.1  TECHNICAL LEMMA FOR THE TEACHER MODEL

**Lemma B.1.** *Let $f_{\mathsf{TE}} = \phi_{\mathsf{TE}}(x)^\top \boldsymbol{w}_{\mathsf{TE}}$, where $\phi_{\mathsf{TE},i}(x) = \sigma(u_{\mathsf{TE},i}^\top x)$ for $i = 1, \ldots, M_{\mathsf{TE}}$. The teacher model weight $\boldsymbol{w}_{\mathsf{TE}}$ has explicit solution form:*

$$\boldsymbol{w}_{\mathsf{TE}} = (\Phi \Lambda \Phi^\top)^{-1} \Phi \Lambda \boldsymbol{w}^*,$$

*where $\Phi_{j,i} = \frac{a_i(u_{\mathsf{TE},j})}{\sqrt{\lambda_i}}$ for $j \in [M_{\mathsf{TE}}], i \in [M^*]$.*

*Proof of Lemma B.1.* Recall that $f_{\mathsf{TE}}(x) = \sum_{i=1}^{M_{\mathsf{TE}}} w_{\mathsf{TE},i} \sigma(u_{\mathsf{TE},i}^\top x)$, and the teacher model weight is defined as:

$$\boldsymbol{w}_{\mathsf{TE}} = \arg\min_{\boldsymbol{w}} \ \mathbb{E}_{x \sim \mathcal{D}} \big\| f^*(x) - \sum_{i=1}^{M_{\mathsf{TE}}} w_i \sigma(u_{\mathsf{TE},i}^\top x) \big\|^2, \tag{37}$$

where $f^*(x) = \sum_{i=1}^{M^*} v_i^* e_i(x)$ with $v^* = \Lambda^{1/2} \boldsymbol{w}^*$. The ground truth covariance is:

$$\Lambda = \mathrm{diag}(\lambda_1, \ldots, \lambda_{M^*}), \quad v^* = \Lambda^{1/2} \boldsymbol{w}^*.$$

Expanding the ReLU function in the spherical–harmonic basis $\{e_i(\cdot)\}_{i=1}^{M^*}$: given $u$, according to the Gaussian Universality assumption (Simon et al., 2023; Medvedev et al., 2025), we have for $x \sim \mathcal{D}$

$$\sigma(u^\top x) = \sum_{\ell=1}^{M^*} a_\ell(u) \, e_\ell(x), \quad \big\{ a_\ell(u) := \langle \sigma(u^\top x), e_\ell(x) \rangle_{\mathcal{D}} \big\}_{i=1}^{M^*} \sim \mathcal{N}(0, \Lambda). \tag{38}$$

We define the projection matrix $\Phi \in \mathbb{R}^{M_{\mathsf{TE}} \times M^*}$ as

$$\Phi_{ji} := \frac{a_i(u_{\mathsf{TE},j})}{\sqrt{\lambda_i}}, \quad j \in [M_{\mathsf{TE}}], i \in [M^*].$$

Note that a fixed teacher neuron $j$ can be expressed as $\sigma(u_{\mathsf{TE},j}^\top x) = \sum_{i=1}^{M^*} a_i(u_{\mathsf{TE},j}) \, e_i(x) = \sum_{i=1}^{M^*} \left(\Phi_{ji} \sqrt{\lambda_i}\right) e_i(x) = \Phi_{j,:}^\top \Lambda^{1/2} e(x)$, where $\Phi_{j,:}$ denotes the $j$-th row of $\Phi$. It means that $\phi_{\mathsf{TE}}(x) = \Phi \Lambda^{1/2} e(x)$. In addition, we rewrite the groundtruth as

$$f^*(x) = \sum_{i=1}^{M^*} v_i^* e_i(x) = \sum_{i=1}^{M^*} w_i^* \sqrt{\lambda_i} \, e_i(x) := \phi^*(x)^\top \boldsymbol{w}^*, \quad \phi^*(x) := \Lambda^{1/2} e(x).$$

Then consider the quadratic loss function:

$$\mathcal{L}(\boldsymbol{w}_{\mathsf{TE}}) = \mathbb{E}_{x \sim \mathcal{D}} \left[ \left( \phi_{\mathsf{TE}}(x)^\top \boldsymbol{w}_{\mathsf{TE}} - \phi^*(x)^\top \boldsymbol{w}^* \right)^2 \right].$$

Because the derivative of the quadratic loss is given as

$$\nabla \mathcal{L}(\boldsymbol{w}_{\mathsf{TE}}) = 2 \mathbb{E}_{x \sim \mathcal{D}} \left[ \phi_{\mathsf{TE}}(x) \phi_{\mathsf{TE}}(x)^\top \right] \boldsymbol{w}_{\mathsf{TE}} - 2 \mathbb{E}_{x \sim \mathcal{D}} \left[ \phi_{\mathsf{TE}}(x) \phi^*(x)^\top \right] \boldsymbol{w}^*$$
$$= 2 \Phi \Lambda \Phi^\top \boldsymbol{w}_{\mathsf{TE}} - 2 \Phi \Lambda \boldsymbol{w}^*.$$

Setting the gradient to zero, we have $\boldsymbol{w}_{\mathsf{TE}} = (\Phi \Lambda \Phi^\top)^{-1} \Phi \Lambda \boldsymbol{w}^*$. $\square$

## B.2 Proof of Theorem 4.5

*Proof.* Follow the notation and proof of Lemma B.1, $\boldsymbol{w}_{\mathsf{TE}}$ has closed form:

$$\boldsymbol{w}_{\mathsf{TE}} = (\Phi\Lambda\Phi^\top)^{-1}\Phi\Lambda\boldsymbol{w}^*.$$

Define

$$Z := \Phi^\top(\Phi\Lambda\Phi^\top)^{-1}\Phi \in \mathbb{R}^{M^* \times M^*}.$$

Using Theorem 4.1 of LeJeune et al. (2024), we know

$$Z \sim (\Lambda + \kappa_{\mathsf{TE}}I)^{-1} = \operatorname{diag}\left\{\frac{1}{\lambda_1 + \kappa_{\mathsf{TE}}}, \dots, \frac{1}{\lambda_{M^*} + \kappa_{\mathsf{TE}}}\right\}. \tag{39}$$

Therefore, we have

$$v_{\mathsf{TE}} = \Lambda^{1/2}Z\Lambda^{1/2}v^* = \frac{\lambda_i}{\lambda_i + \kappa_{\mathsf{TE}}}v_i^* = L_i v_i^*.$$

Hence the teacher prediction is given by

$$v^* = \Lambda^{1/2}\boldsymbol{w}^*, \quad v_{\mathsf{TE}} = \phi_{\mathsf{TE}}(x)^\top\boldsymbol{w}_{\mathsf{TE}} = \Lambda^{1/2}\Phi^\top(\Phi\Lambda\Phi^\top)^{-1}\Phi\Lambda\boldsymbol{w}^* = \Lambda^{1/2}Z\Lambda^{1/2}v^*.$$

Then use Lemma A.1 proof, under the substitutions $d \mapsto M^*$, $\Lambda \mapsto \Lambda$, $U_{\mathsf{TE}} \mapsto \Phi$, $\boldsymbol{Z} \mapsto Z$, the MSE of ReLU teacher model is given as

$$\mathcal{E}_{\mathsf{TE}} = \frac{M_{\mathsf{TE}}}{M_{\mathsf{TE}} - \sum_{j\in[M^*]} L_j^2} \sum_{i\in[M^*]} (1 - L_i)^2 (v_i^*)^2.$$

In addition, following the same proof of Theorem 4.4 yields weak-to-strong ratio lower bound:

$$\frac{\mathcal{E}_{\mathsf{st}}(T)}{\mathcal{E}_{\mathsf{TE}}} \geq \sum_{i\in[M^*]} \left(\frac{\mathcal{E}_{\mathsf{TE}}}{v_i^{*2}} + M_{\mathsf{TE}}\right)^{-1}.$$

$\square$

## B.3 Proof of Theorem 4.6

**Theorem B.2** (Restatement of Theorem 4.6). *Consider the groundtruth setting: let $r \in (1,3), \alpha > 1, \beta < \alpha - 1/2$ and $k_{st} = k^* = k_{te}^r$, $k_{te} \gg d = 4$, $w_1^* = m^\alpha$, $w_i^* = 0$ for $2 \leq i \leq N_{k^*-1}$, and $w_j^* = m^\beta$ for $N_{k^*-1} < j \leq N_{k^*}$, where $m := M_{\mathsf{TE}} = \sum_{k=0}^{k_{te}} N_k = \Theta(k_{te}^{d-1})$ and $M^* = m^r = \Theta(k^{*(d-1)})$. Then weak-to-strong ratio satisfies $\mathcal{E}_{\mathsf{st}}(T)/\mathcal{E}_{\mathsf{TE}} \leq \Theta(m^{-1}) + \Theta(m^{2\beta - \frac{r}{3} - 2\alpha + 2})$.*

*Proof.* Lemma C.3 yields

$$\sigma_k = \Theta(k^{-\frac{d+2}{2}}), \quad \sigma_1 = \frac{1}{2d}, \quad N_k = \Theta(k^{d-2}), \quad k^* = m^{\frac{r}{d-1}}, \quad k_{\mathsf{TE}} = m^{\frac{1}{d-1}}.$$

Applying the upper bound in Lemma C.2 with $k = 1$, $\kappa_{\mathsf{TE}}$ has the following upper bound:

$$\kappa_{\mathsf{TE}} \leq \frac{\sum_{i=1}^{M^*} \lambda_i}{M_{\mathsf{TE}} - 1} = \frac{1/2d + k^{*(-3)}}{M_{\mathsf{TE}} - 1} \leq \frac{1/2d + m^{\frac{-3r}{d-1}}}{M_{\mathsf{TE}} - 1} = \Theta(m^{-1}).$$

Applying the lower bound in Lemma C.2 with $k = 1$, $\kappa_{\mathsf{TE}}$ has the following lower bound:

$$\kappa_{\mathsf{TE}} \geq \lambda_{M^*}\left(\frac{M^*}{M_{\mathsf{TE}}} - 1\right) = m^{\frac{-r(d+2)}{2(d-1)}}\left(\frac{m^r}{M_{\mathsf{TE}}} - 1\right) = \Theta(m^{\frac{r(d-4)}{2(d-1)} - 1}) = \Theta(m^{-1}).$$

The upper and lower bounds imply that $\kappa_{\mathsf{TE}} = \Theta(m^{-1}) = o(1)$ as $m \to \infty$.
Recall the definition of $L_i$:

$$L_i = \frac{\lambda_i}{\lambda_i + \kappa_{\mathsf{TE}}} = \begin{cases} 1/(1 + 2dm^{-1}) & i = 1 \\ \vdots & \\ m^{-r}/(m^{-r} + m^{-1}) = 1/(1 + m^{r-1}) & N_{k^*-1} < i \leq N_{k^*}, \end{cases}$$

and note that $N_{k^*-1} \leq i \leq N_{k^*}$, $v_i^* = \sigma_i w_i^* = m^{\beta - r/2}$, the MSE of teacher model in Theorem 4.1 (which implies that $N_{k^*} = m^{2r/3}$, ), and $2\alpha + \frac{r}{3} - 2\beta - 2 > 0$ (by the choice of $r, \alpha, \beta$ in the statement of the Theorem), we have the following:

$$\mathcal{E}_{\mathsf{TE}} = \frac{1}{1 - \frac{1}{M_{\mathsf{TE}}}\sum_i L_i^2} \left( (1 - L_1)^2(v_1^*)^2 + \sum_{i \geq N_{k^*-1}}^{N_{k^*}} (1 - L_i)^2(v_i^*)^2 \right)$$

$$= \frac{1}{1 - \Theta(m^{r/3-1})} \left( m^{2\alpha-2} + \frac{m^{2r/3}m^{2(r-1)}m^{2\beta-r}}{(1 + m^{r-1})^2} \right)$$

$$\overset{(i)}{=} \Theta(m^{2\alpha-2}) + \Theta(m^{-\frac{r}{3}+2\beta}) = \Theta(m^{2\alpha-2}), \tag{40}$$

where (i) follows from $\kappa_{\mathsf{TE}} = \Theta(m^{-1})$ and $2\alpha + \frac{r}{3} - 2\beta - 2 > 0$.

We can choose the time $T$ such that $\gamma_1(T) = 1 - e^{-2\hat{\lambda}_1 T} = \frac{1}{\frac{\mathcal{E}_{\mathsf{TE}}}{m^{2\alpha+1}}+1} = \frac{1}{1+m^{-3}}$. Therefore, we have $T = \Theta(\log m)$. Noting that $L_i = \Theta(m^{1-r})$, we know that for all $i$ satisfying $N_{k^*-1} < i \leq N_{k^*}$, the following holds:

$$\gamma_i(T) = 1 - e^{-2T/m^r} \leq \Theta(m^{-r}\log m),$$

where we used inequality $1 - e^{-x} \leq x$.

So $\gamma_i(T)$ has the following upper bound:

$$\gamma_i(T) \leq \frac{m^{r-1}}{\frac{\mathcal{E}_{TE}}{m^{1+2\beta-r}}+1} = \frac{1}{L_i(\frac{\mathcal{E}_{TE}}{m\langle f^*, e_i\rangle^2}+1)} = \frac{B_i}{A_i}.$$

Thus at time $T$, we have

$$\sum_{i>1}^{M^*} A_i\left(\gamma_i(T) - \frac{B_i}{A_i}\right)^2 + (v_i^*)^2 - \frac{B_i^2}{A_i} \leq \sum_{i>1}^{M^*} A_i\left(\gamma_i(0) - \frac{B_i}{A_i}\right)^2 + (v_i^*)^2 - \frac{B_i^2}{A_i} \leq \sum_{i>1}^{M^*}(v_i^*)^2, \tag{41}$$

where we used $\gamma_i(0) = 0$ and $\gamma_i(t)$ is increasing over $t$.

Moreover, we have

$$\frac{B_1^2}{A_1} = \frac{L_1^2(v_1^*)^4}{\frac{L_1^2\mathcal{E}_{\mathsf{TE}}}{m} + L_1^2(v_1^*)^2} = \frac{m^{4\alpha}}{\mathcal{E}_{\mathsf{TE}}/m + m^{2\alpha}} = \frac{m^{2\alpha}}{m^{-3}+1}, \tag{42}$$

where we used $\mathcal{E}_{\mathsf{TE}} = \Theta(m^{2\alpha-2})$. Therefore, we have

$$\mathrm{v}_1^{*2} - \frac{B_1^2}{A_1} = m^{2\alpha} - \frac{m^{2\alpha}}{m^{-3}+1} = \Theta(m^{2\alpha-3}). \tag{43}$$

Now we are ready for deriving the MSE formula for the student.

From Eq. 33, the MSE of the student model at time $T$ satisfies

$$\mathcal{E}_{\mathsf{st}}(T) = \sum_{i\in[M^*]} A_i\left(\gamma_i(T) - \frac{B_i}{A_i}\right)^2 + (v_i^*)^2 - \frac{B_i^2}{A_i}$$

$$= (v_1^*)^2 - \frac{B_1^2}{A_1} + \sum_{i>1} A_i\left(\gamma_i(T) - \frac{B_i}{A_i}\right)^2 + (v_i^*)^2 - \frac{B_i^2}{A_i}$$

$$\overset{(i)}{\leq} (v_1^*)^2 - \frac{B_1^2}{A_1} + \sum_{i>1}^{M^*}(v_i^*)^2$$

$$\overset{(ii)}{=} \Theta(m^{2\alpha-3}) + \Theta\left(m^{2\beta-\frac{r}{3}}\right), \tag{44}$$

where (i) follows from (41), (ii) follows from (33). Using $\mathcal{E}_{\mathsf{TE}} = \Theta(m^{2\alpha-2})$ again, we can upper bound the ratio as:

$$\frac{\mathcal{E}_{\mathsf{st}}(T)}{\mathcal{E}_{\mathsf{TE}}} \leq \Theta(m^{-1}) + \Theta(m^{2\beta-\frac{r}{3}-2\alpha+2}). \tag{45}$$

$\square$

## C  AUXILIARY LEMMAS

**Lemma C.1** (Bai-Yin's law, see Bai et al. (1993)). *Let* $\mathbf{A}$ *be an* $N \times n$ *random matrix whose entries are independent copies of a random variable with **zero mean, unit variance, and finite fourth moment**. Suppose that the dimensions* $N$ *and* $n$ *grow to infinity while the aspect ratio* $n/N$ *converges to a constant in* $[0, 1]$. *Then*

$$s_{\min}(\mathbf{A}) = \sqrt{N} - \sqrt{n} + o(\sqrt{n}), \quad s_{\max}(\mathbf{A}) = \sqrt{N} + \sqrt{n} + o(\sqrt{n}), \quad \textit{almost surely,}$$

*where* $s_{\min}(\cdot)$ *and* $s_{\max}(\cdot)$ *denotes the minimum and maximum singular value, resceptively.*

**Lemma C.2** (Lemma I.1 from Simon et al. (2023)). *The quantity* $\kappa_{\mathsf{TE}}$ *satisfies that*

$$\kappa_{\mathsf{TE}} \leq \frac{\sum_{i>k} \lambda_i}{M_{\mathsf{TE}} - k}, \quad \forall k \in \{1, \ldots, M_{\mathsf{TE}}\},$$

$$\kappa_{\mathsf{TE}} \geq \lambda_k \left( \frac{k}{M_{\mathsf{TE}}} - 1 \right), \quad \forall k \in \{M_{\mathsf{TE}}, \ldots, d\}.$$

*Specifically, if* $\lambda_i \simeq i^{-\alpha}$ *with* $\alpha > 1$ *for* $i \in [d]$, *then* $\kappa_{\mathsf{TE}} = \Theta\left(M_{\mathsf{TE}}^{-\alpha}\right)$.

**Lemma C.3** (Lemma E.8 and Corollary E.9 in Medvedev et al. (2025)). *Depending on the size of* $d$ *and* $k$, *the following holds for the size of coefficients of ReLU activation in the basis of Legendre polynomials:*

$$\sigma_k = \frac{\Gamma\left(\frac{d}{2}\right)}{\sqrt{\pi}\, \Gamma\left(\frac{d-1}{2}\right)} \cdot \frac{1}{2^k} \cdot \frac{1}{\left(\frac{k}{2} + \frac{d-1}{2}\right) \cdots \left(\frac{d-1}{2}\right)} \quad \textit{for even } k,$$

$$\sigma_1 = \frac{2}{d}, \quad \textit{and} \quad \sigma_k = 0 \textit{ for odd } k > 1.$$

*Asymptotically, this implies:*

*1.* $\sigma_0 = \sqrt{\frac{2}{\pi d}} + \Theta(d^{-3/2}), \quad \sigma_1 = \frac{1}{2d}$.

*2. If* $k = \Theta(1)$, *then for even* $k$, $\sigma_k = \Theta\left(d^{-\frac{k+1}{2}}\right)$.

*3. If* $k \gg d$, *then for even* $k$, $\sigma_k = \Theta\left(k^{-\frac{d-1}{2} - \frac{3}{2}}\right)$.

*The following bounds hold for* $\sigma_k^2 N_k$ *for* $k$ *even or* $k = 1$:

$$\begin{cases} \sigma_k^2 N_k = \Theta(k^{-4}), & \textit{if } k \gg d, \\ \sigma_k^2 N_k = \Theta(d^{-1}), & \textit{if } k \ll d, \\ \sigma_k^2 N_k = 0 & \textit{if } k > 1 \textit{ is odd.} \end{cases}$$

**Lemma C.4** (Proposition B.3 in Medvedev et al. (2025)). *The Diagonal Covariate Feature model* $f = \sum_{i=1}^{\infty} \langle f, e_i \rangle_{\mathcal{D}} e_i$, *where* $\{e_i\}_{i=1}^{\infty}$ *is the spherical harmonics* $\{\{\phi_{i,k}\}_{i=1}^{N_k}\}_{k=1}^{\infty}$, *and* $\{\langle f, e_i \rangle_{\mathcal{D}}\}_{i=1}^{\infty}$ *is drawn from:*

$$\{\langle f, e_i \rangle_{\mathcal{D}}\}_i^{\infty} \sim \mathcal{N}(0, \Lambda), \quad \textit{with } \Lambda = (\underbrace{\sigma_1, \ldots, \sigma_1}_{N_1 \textit{ times}}, \underbrace{\sigma_2, \ldots, \sigma_2}_{N_2 \textit{ times}}, \ldots, \underbrace{\sigma_k, \ldots, \sigma_k}_{N_k \textit{ times}}),$$

*where* $\sigma_k$ *is repeated* $N_k$ *times in decreasing order. Furthermore,* $\sigma_k$ *is nonzero only for even* $k$ *and* $k = 1$.

*This is equivalent to the ReLU model with uniformly distributed first-layer weights* $u$ *and uniformly distributed input* $x$, *i.e.,*

$$f(x) = \sum_{i=1}^{M} w_i\, \sigma(u_i^{\top} x), \quad u, x \sim \mathrm{Unif}(\mathbb{S}^{d-1}),$$

*where* $M = \sum_{i=1}^{k} N_i$.

Table 1: Student model test accuracy under different teacher capacities.

| $M_{\mathsf{TE}}$ | Student Test Accuracy (%) |
|---|---|
| 50 | 96.15 |
| 75 | 96.64 |
| 100 | 97.12 |

Table 2: Model sizes for simulating Theorem 4.3 (Section 5.1). $M_{\mathsf{TE}}$ is the number of learnable teacher parameters, $d$ is the number of parameters in the ground-truth model, and $M_{\mathsf{ST}}$ is the number of learnable student parameters. For the experiments in Figure 1, $d$ and $M_{\mathsf{ST}}$ are computed as functions of $M_{\mathsf{TE}}$ according to the statement of Theorem 4.3.

| $M_{\mathsf{TE}}$ | 4 | 8 | 16 | 32 | 64 | 128 |
|---|---|---|---|---|---|---|
| $d$ | 8 | 23 | 64 | 181 | 512 | 1448 |
| $M_{\mathsf{ST}}$ | 13 | 50 | 181 | 664 | 2435 | 8932 |

## D  ADDITIONAL EXPERIMENTAL RESULTS

**Real World Data Experiments.** We conduct additional experiments on the MNIST dataset, use the cross-entropy loss and the Adam optimizer with learning rate $\eta = 0.001$, and average the results over 5 independent runs. To demonstrate the weak-to-strong generalization (W2SG) phenomenon, we trained a two-layer ReLU network with varying hidden sizes, where both layers are trained simultaneously. Specifically, we varied the teacher model size $M_{\mathsf{TE}} = \{50, 75, 100\}$, while fixing the student size $M_{\mathsf{ST}} = 300$.

Table 1 reports the mean training and test losses, along with test accuracies for student models trained under different teacher capacities.

The results in Table 2 demonstrate that as the teacher model capacity increases, the student achieves improved test performance, thereby validating the W2SG phenomenon in real-world settings. This observation is consistent with the theoretical insights of Theorem 4.6.

## E  THE USE OF LARGE LANGUAGE MODELS (LLMS)

LLMs are not involved in our research methodology or analysis. Their use is limited to polish the writing.

