# OpenReview forum: "Provable Weak-to-Strong Generalization via Overspecified Students and Underspecified Teachers"
_ICLR.cc/2026/Conference — Submitted to ICLR 2026_

### Official Review · Reviewer_dRfG · 2025-10-16

**Soundness:** 2
**Presentation:** 1
**Contribution:** 2
**Rating:** 2
**Confidence:** 4

**Summary:**

This paper investigates Weak-to-Strong Generalization (W2SG), the phenomenon where a powerful student model, trained solely on labels generated by a weak teacher model, can still surpass the teacher's performance. The authors establish a model within a controlled theoretical framework (using two-layer neural networks with random features), where the teacher model is set as underspecified (unable to fully learn the ground-truth function), and the student model is overspecified (capable of precisely recovering the ground-truth function). The core finding is that the weak teacher, due to limited capacity, generates inherent errors in the low-variance directions of the data covariance matrix. Conversely, the student model, trained via gradient flow, exhibits an implicit bias: its convergence is slower in these low-variance directions. Consequently, the student can use early stopping to explicitly avoid learning the teacher's errors in these directions, thereby achieving superior generalization performance. The paper also derives the spectral conditions under which W2SG occurs.

**Strengths:**

* The paper provides a provable theoretical mechanism for the W2SG phenomenon under a realistic "underspecified teacher" setting.
* It offers an analysis of the training dynamics using the Eigenlearning framework, revealing the crucial role of implicit bias in filtering out teacher errors.
* It derives explicit closed-form expressions for the prediction errors and provides precise spectral conditions for W2SG occurrence.

**Weaknesses:**

First, please feel free to correct me if I've misunderstood any points. **If my concern is satisfactorily addressed, I will certainly increase the score.**

## 1. Problem Setting
I do not believe the "Overspecified Students and Underspecified Teachers" setting in this paper is a unique advantage; rather, it is the defining characteristic of the original W2SG problem. Upon seeing the title, "Provable Weak-to-Strong Generalization via Overspecified Students and Underspecified Teachers," a natural question arises: Is this not precisely what W2SG is about? I understand that some existing works simplify W2SG to a weak-supervised learning framework by, for example, using mask mechanisms (Ildiz et al. 2025) or misfit-based analyses (Mulgund & Pabbaraju 2025) that abstract away the *strong* versus *weak* distinction. While these offer interesting insights, this paper uses parameter count to define the strength difference. More concerningly, the authors assume the strong student model has more parameters than the ground truth function itself. This is an unusual setting in practice, as the strong student's goal is to recover the ground truth, not exceed its complexity. I encourage the authors to discuss the motivation and practical relevance of this specific parameter-based setting.

## 2. Related Work
### Citation Format
To my best knowledge, many of the citations in this submission are informal and should be revised. The original paper Burns et al. (2023) has been accepted to ICML 2024. Similarly, many theoretical papers about W2SG (e.g., Medvedev et al. (2025), Ildiz et al. (2025), Mulgund & Pabbaraju (2025), Dong et al. (2025), and Wu & Sahai (2025)) have been accepted by either ICLR 2025 or ICML 2025. Please update all citations accordingly.

### Missing Theoretical Context
The paper overlooks several key theoretical works on W2SG, leading to a lack of necessary discussion on how this work relates to previous theory. For instance, the representation-based analysis [1] and feature learning-based theories [2-3] are highly relevant, especially since [2-3] also use random matrix theory, which is central to your paper. Given your use of a two-layer neural network and eigenvalue analysis, some of your conclusions may overlap with prior findings.

* When discussing how large/small eigenvalues influence W2SG in specific positions, you should cite [4] and provide a comparative analysis, as [4] also analyzes the impact of eigenvalue scales.
* When discussing the role of low-variance directions, you should cite [5] and discuss how your ideas support or differ from theirs, especially since both works use random matrix theory.
* Additionally, both Mulgund & Pabbaraju (2025) and [6-7] generalized the result of Charikar et al. (2024) to a general class of loss functions, a point that should be acknowledged in the related work section.


[1] Representations Shape Weak-to-Strong Generalization: Theoretical Insights and Empirical Predictions. ICML 2025.

[2] From Linear to Nonlinear: Provable Weak-to-Strong Generalization through Feature Learning. High-dimensional Learning Dynamics 2025.

[3] On the Mechanisms of Weak-to-Strong Generalization:
A Theoretical Perspective. arXiv:2505.18346.

[4] High-dimensional Analysis of Knowledge Distillation: Weak-to-Strong Generalization and Scaling Laws. ICLR 2025.

[5] Discrepancies are Virtue: Weak-to-Strong Generalization through Lens of Intrinsic Dimension. ICML 2025.

[6] Revisiting Weak-to-Strong Generalization in Theory and Practice: Reverse KL vs. Forward KL. Findings of ACL 2025.

[7] On the Emergence of Weak-to-Strong Generalization: A Bias-Variance Perspective. arXiv:2505.24313.

## 3. Theory

### Theorem 4.1
There is a lack of discussion to help readers understand the bounds. Remark 1 states that,
> ...when the teacher is underspecified (i.e., $M_{\mathrm{TE}}<d$), there must exist some directions $\boldsymbol{e}_i$ for which $L_i$ is significantly smaller than 1, indicating that the teacher fails to fully learn the corresponding components of the ground truth.

Why "must exist" directions where $L_i$ is "significantly smaller" than 1? The meaning of "significantly smaller" needs to be rigorously defined or discussed. You must substantiate your claims with sufficient discussion to prevent reader confusion.

While Remark 2 notes,
> By analyzing the teacher error expression in (5), we can show that the teacher's MSE can become large when the eigenvalue distribution of the data covariance matrix $\Lambda$ is clustered...

I agree with this statement, but the authors do not discuss the conditions under which the eigenvalue distribution of the data covariance matrix $\Lambda$ becomes "clustered" in practice. Furthermore, you may discuss how this clustering condition impacts the occurrence of W2SG.

### Theorem 4.2
This theorem, in its current presentation, seems insufficient to verify the mechanism of early stopping. A smaller time $t$ makes $\gamma_i(t)=\left(1-e^{-\lambda_i t}\right) L_i \to 0$. However, the student error $\mathcal{E}_{\mathrm{st}}(t)$ depends on $\left(\gamma_i(t)-\frac{B_i}{A_i}\right)^2$. A smaller $t$ does not automatically guarantee a smaller quadratic form. You can explicitly explain whether $\gamma_i(t)$ is larger or smaller than $\frac{B_i}{A_i}$ for the low-variance directions ($\lambda_i$) at the optimal stopping time $t$, or incorporate more discussion on this crucial point without requiring the reader to consult the appendix.

### Theorem 4.3
The theorem makes a very strong and specific assumption about the spectral properties, particularly the eigenvalue distribution. Let's review the statement:
> ...$\lambda_i=\Theta(1)$ if $i \leq K$, and $\lambda_i=\Theta\left(m^{-1}\right)$ for $K<i \leq d$, and choose the ground truth weight sush that $\lambda_i\left(\theta_i^*\right)^2=\Theta\left(m^\alpha\right)$ if $i \leq K$ and $\lambda_i\left(\theta_i^*\right)^2=\Theta\left(m^{-\beta}\right)$. If $\alpha \geq 1, \beta>0$ and $\alpha+\beta \in(2,3)$...

How does such a strong assumption about the eigenvalue and ground truth weight distribution apply in practice? You should cite well-established papers to support the plausibility of this assumption or provide extensive explanation. Without this, the applicability of the theory is unclear. Furthermore, the condition $\alpha+\beta \in(2,3)$ is a critical requirement for the theorem to hold. The authors should discuss the practical mechanisms or conditions that guarantee $\alpha+\beta \in(2,3)$ and elaborate on the interplay between $\alpha$ and $\beta$.

### Theorem 4.4
The accompanying discussion in Remark 5 seems insufficient and may be misleading. Theorem 4.4 provides a lower bound for the MSE-ratio. Remark 5 states:
> ...the MSE-ratio becomes smaller if we increase the teacher model size.

While increasing the teacher model size reduces the *lower bound* of the MSE-ratio, a smaller lower bound does not imply a smaller MSE-ratio. Instead, a tighter upper bound is what would guarantee a smaller MSE-ratio. The authors should discuss this distinction more carefully.

### Theorem 4.6
The full statement of Theorem 4.6 contains numerous variables that require clearer definition and context. The provided Remark 7 offers only a very short explanation:
> Remark 7: This setting aligns the ground-truth expansion and the eigenvalue ordering with the ReLU kernel’s spherical–harmonic decomposition.

This is highly confusing and makes the theory very difficult to grasp. You should provide a detailed and intuitive explanation of what each variable represents and how they relate to W2SG.

## 4. Experiments
The authors validate their theory using a two-layer NN, with MNIST as the largest dataset. However, almost all previous theoretical papers on W2SG, including the original work, conduct experiments on NLP tasks, which are the natural domain for W2SG concerning LLMs. While I acknowledge the authors may be following the experimental setting of papers like (Medvedev et al., 2025) and may face difficulties conducting LLM-scale experiments, the current experimental setup does not clearly demonstrate how this theory can inspire or guide practitioners working with W2SG in the LLM context. The authors should elaborate on the translational value of their current experiments to real-world W2SG problems.

**Questions:**

- Could you provide a clearer practical and theoretical motivation for assuming the strong student model has more parameters than the ground-truth function itself?
- What is "significantly smaller than $1$" in Remark 1?
- How does Theorem 4.2 explicitly verify the early stopping mechanism?
- What are the practical mechanisms or conditions that guarantee $\alpha+\beta \in(2,3)$, and what is the intuitive interplay between $\alpha$ and $\beta$?
- How to intuitively understand all variables in Theorem 4.6 and how they quantitatively relate to the W2SG phenomenon?

---

### Official Review · Reviewer_oKHG · 2025-10-29

**Soundness:** 1
**Presentation:** 2
**Contribution:** 1
**Rating:** 2
**Confidence:** 4

**Summary:**

This paper theoretically investigates weak-to-strong generalization using the framework of random feature models. The analysis focus on a setting with an underspecified teacher (weak) model (small width) and an overspecified (strong) model (large width). The key contributions are provable guarantees for weak-to-strong generalization in two specific scenarios: (1) linear networks with linear targets and (2) ReLU networks with polynomial targets.

**Strengths:**

This work provides rigorous theoretical guarantees for weak-to-strong generalization, a phenomenon that has recently gained considerable attention from both the practical and theoretical research communities.

**Weaknesses:**

My main concern is the insufficient comparison to the closely related work of Medvedev et al. (2025). Although the authors mention this paper (lines 66-67, 108), they don't clearly explain the differences between the problem setup and the results of the two papers. A more detailed comparison is needed so readers can understand what is truly new in this work. (See Questions for my detailed questions/concerns).

---
[1] Medvedev et al. Weak-to-Strong Generalization Even in Random Feature Networks, Provably. ICML 2025.

**Questions:**

* The paper states (lines 66-67) that Medvedev et al. (2025) consider the setting where both teacher and student are overspecified. I am not sure if this characterization is fully accurate. My reading of Medvedev et al. (2025), specifically Theorem 3.2 of their work (for the linear network/target case), seems to analyze a setting with an underspecified teacher ($M_{TE} = (d-k)^{2/3}$).
* Could the authors provide a detailed, side-by-side comparison of Theorem 4.3 with Theorem 3.2 (and its full version Theorem D.5) in Medvedev et al. (2025)? In my reading, it seems that translating problem parameter orders and applying slight modifications to the results in Medvedev et al. (2025) might lead to the result in Theorem 4.3. If I am wrong, it would be beneficial for the authors to explicitly highlight the key technical differences and novelties of their result (and ideally, add this clarification to the main draft as well).
* What is the meaning of "We assume that the teacher model $f_{TE}$ and the student model $f_{st}$ are even polynomials of degree $k_{TE}$ and $k_{st}$, respectively" in line 287-288? In addition, the overall "Groundtruth Model" paragraph (lines 281-289) is somewhat difficult to parse and should be polished for clarity.
* Does the teacher model considered in this work for ReLU network case is overspecified? It appears that satisfying the condition in Theorem 4.6, requires $M_{TE}$ to be much larger than dimension $d$, which implies overspecified model. Could the authors provide a detailed comparison between this result (Theorem 4.6) and the corresponding result in Medvedev et al. (2025), specifically Theorem 6.5?
* Given the significant overlap in problem settings, it would be extremely helpful if the authors explicitly highlighted what new analytical techniques or proof strategies, if any, were developed in this work that differ from or extend those in Medvedev et al. (2025).

---

### Official Review · Reviewer_DZyZ · 2025-10-31

**Soundness:** 2
**Presentation:** 2
**Contribution:** 1
**Rating:** 2
**Confidence:** 5

**Summary:**

The paper considers a weak-to-strong generalization setup from Medvedev et al 2025 with two-layer random feature networks with fixed random first layer and trainable second layer trained with gradient flow on population, and the student is trained with early stopping. For the ground truth, they take, as in Medvedev et al 2025, either a linear network with Gaussian data having a diagonal covariance structure or a ReLU network with isotropic data. Their main contribution is that they provide, conditional on Gaussian Universaliy Ansatz (i.e. using the Eigenframework), show (i) closed form expression for prediction error of student and teacher, (ii) the sufficient conditions on the data covariance under which weak-to-strong generalization occurs, and (iii) experiments exploring the dependence on the ratio of student and teacher error on the relative size of the two networks. Some of the claimed contributions are mischaracterized: (i) the first two contributions claimed are already written in Medvedev et al 2025 (see Weaknesses) (ii) the authors say that under the conditions found weak-to-strong generalization provably occurs  (iii) the authors claim that the larger teacher leads to stronger weak-to-strong generalization - I think this is not entirely true (see Weaknesses).

**Strengths:**

Exploring what the Eigenframework and Gaussian Universality Ansatz can give us in the context of Weak-to-Strong generalization is a great direction which has not been previously explored in the literature, so despite the paper following up and being based on a lot of things already explored in Medvedev et al 2025, it is definitely a novel contribution and one worth exploring. However, some of the contributions claimed in the paper have already been known from Medvedeve et al 2025.


The main insight from the Eigenframework is the availability of the closed form for both student and teacher losses, which circumvents the difficulty of lower bounding the teacher error in weak-to-strong setup and the authors deserve credit for this observation. It does come with the price of relying on Gaussian Universality Ansatz.


Some of the results and writing are not completely clear or correct and could use another round of polishing. Writing is generally good and clear but occasionally authors misrepresent results from the literature or are not precise enough in their claims.


The significant contributions are:
1. A new approach to considering w2s generalization using the Eigenframework
2. Analysis of the problem setup using the eigenframework and the derivation of necessary conditions on the covariance for establishing w2s generalization under GUA
3. Experiments exploring the ratio of w2s improvement as a function the relative sizes of student and teacher

**Weaknesses:**

1. This paper is completely based on the Eigenlearning framework (Simon et al 2023), which is $\textbf{informal}$ and only rigorously proved in very specific scenarios (not in any fixed dimension; only when the dimension increases with sample size e.g. see Misiakiewicz and Saeed 2024.) It can be formalized conditional on the Gaussian Universality Ansatz (GUA), which states (from Medvedev et al. 2025) “The Gaussian Universality Ansatz states that when sampling x the Gaussian universality holds for the eigenfunctions in the sense that the expected risk remains unchanged if we replace them with Gaussian with appropriate parameters.“ This certainly is unproven for the ReLU network case (Section 4.2) and I’m unsure if it holds for linear networks (since the requirement for GUA is on the eigenfunctions). I think this at least deserves some discussion in the main body (or even intro/when talking about contributions).

2. The first two contributions that the teacher exhibits large prediction error in low variance directions and the implicit bias showed were already known in Medvedev et al 2025. See Section 5 (in version 1): “This makes learning directions $x_i$ with larger $\psi_i$ [variance], i.e. for smaller $i$, significantly faster than the ones with smaller $\psi_i$, i.e. for larger $i$. By appropriately choosing the stopping time, this allows us to effectively zero out the signal $f$ in directions $x_i$. Let us now turn to the teacher. Although the signal is entirely in the first few high energy coordinates of $x$, if we have a small number of teacher nodes $M_{TE}$, they will not be directly represented in the hidden layer, i.e. in $U_{T}x$. Instead, the teacher can only learn $\beta_T \in span(U_T)$, and so learns $\beta_T$ to be the projection of $f^*$ to $span(U_T)$. This means that $\beta_T$ has some energy (non-zero coefficients) in all the coordinates. The student shrinkage will reduce the noise in low energy coordinates by zeroing out the coefficients along those directions. Therefore, theweak-to-strong improvement comes exactly from improvement along the noise directions.” To me this is almost entirely encompassing the two claimed contributions. I think the authors should redo the analysis of related works and more transparently attribute previous contributions to already existing papers. Specifically, since the paper is clearly based on Medvedev et al 2025, I think there should be a more explicit discussion on the novelty of the results compared to what’s already shown in Medvedev et al 2025 (perhaps by clearly stating, in the language of the paper, what’s already shown and what is new).

3. There a number of things that are, in my opinion, incorrectly stated or phrased in the current version of the paper:

   a. Lines 67-69: The teacher model is not overspecified in Medvedev et al 2025 for the case of linear networks. Further, even for ReLU (in both cases) for any finite $M_{TE}$, almost surely the teacher is unable to fit the ground truth because the teacher features do not align with the ground truth basis (i.e. the eigenbasis). It is true that the model is overparametrized in Medvedev et al 2025 Theorem 3.1 which is the ReLU result. But Medvedev et al 2025 show in their Theorem 6.7 the bound on weak-to-strong generalization improvement for other values of $M_{TE}$. The requirement there on $M_{TE}$ is the same as the author’s one in Theorem 4.6: they both establish weak-to-strong generalization when $M_{TE}$ is large in terms of $\sigma_k$ (which the authors took for large $k$ when it’s of the order $k^{d-1}>>d$). Further, even looking at linear networks, in Medvedev et al’s Theorem 3.2 the teacher has size $M_{TE}=(d-k)^{⅔}$, which is the same order compared to $K$ and $d$ as the authors have in their Theorem 4.3 for linear networks. So there is no difference in how over or underparametrized the teacher and student models are.

   b. Lines 92-93: “a larger teacher model size leads to stronger weak-to-strong generalization” mischaracterizes weak-to-strong generalization in this setting. It is true that, as the authors show in e.g. Theorem 4.3, or as was shown in Medvedev et al 2025 Theorems 3.1, 6.7, 6.8, that as $M_{TE}$ increases the ratio of the student to teacher error decreases. But if you look at the absolute size of the decrease of student error vs teacher error, it actually $\textbf{decreases}$ with teacher size increasing. This is also easy to see directly: as $M_{TE}$ increases, the teacher feature space is more aligned with the first $K$ eigendirections (i.e. the support of the ground truth), so the error in low variance directions decreases (where the bulk of w2s improvement comes from). So the absolute size of weak-to-strong improvement has to decrease with teacher size increasing.

   c. When talking about main contributions, e.g. line 90: claiming that the authors show that “weak-to-strong generalization provably occurs” is not true for ReLU case at least, due to dependence on GUA. On the other hand, Medvedev et al 2025’s result on ReLU network (Theorem 3.1 and/or Theorem 6.7) provably establishes that w2s occurs - they use GUA $\textbf{only for the rewriting the result}$ in the form $L_{ST}/L_{TE}<\dots$.

4. Further, and more alarmingly, the paper $\textbf{miscites and uses incorrectly a results}$ for the ReLU section:

   a. In lines 279-290: The author’s result on ReLU networks result (Theorem 4.5 and Theorem 4.6) rests on the use of finite dimensional covariance $\Lambda$ to make the ReLU setting equivalent to the linear network setting, and for that it cites Lemma C.4 which is the Proposition B.4 in Medevedev et al 2025. Their citation of this result is incorrect. As written on lines 1227-1241, Lemma C.4 is actually incorrect (perhaps due to a formatting error in Medvedev et al 2025 (arxiv version 1)). The covariance $\Lambda$ needed there is actually infinitely dimensional - the authors even write “$\{ \langle f, e_i \rangle_{D}\}_{i=1}^{\infty} \sim N(0,\Lambda)$ with $\Lambda = (\sigma_1,\dots,\sigma_k)$” - in the first equation we need infinitely dimensional $\Lambda$ but the $\Lambda$ given is finitely dimensional. In fact, careful reading of Proposition B.4 in Medvedev et al 2025 or its proof makes it clear that the claim is based on Diagonal Feature Covariate Features model defined right above in Proposition B.1 which is over an infinitely dimensional eigenbasis and requires infinitely dimensional $\Lambda$. The rest of section 4.2 on ReLU networks is based on this so I’m not sure if it’s correct. I do think that the Eigenframework actually applies in the infinitely dimensional case, but I’m unsure if all the results transfer because e.g. the teacher model would have it’s error spread over infinitely many directions as opposed to finitely many so the decomposition of teacher error might be different.

**Questions:**

1. Does the Gaussian Universality Ansatz hold for linear networks?
2. How are the the insights about how the teacher predictor looks and the implicit bias of the student different from what was already known in Medvedev et al 2025?
3. Can you clearly state the claim on overspecified/underspecified student and teacher models and how is the size the teacher network in your results different from Medvedev et al 2025?
4. Can you say more about the claim the larger teacher size leads to large w2s improvement? Am I missing something in my characterization of it?
5. Stated briefly, what is the main insight from the experiments?

---

### Official Review · Reviewer_gt2z · 2025-11-01

**Soundness:** 3
**Presentation:** 2
**Contribution:** 1
**Rating:** 4
**Confidence:** 3

**Summary:**

This paper theoretically analyzes weak-to-strong generalization in a regime where the teacher cannot recover the
ground-truth (underspecified), and the student can achieve exact recovery (overspecified). For the analysis, the authors consider the random feature model (two-layer neural networks with random features and trainable linear output layers) for the teacher and student, while the ground-truth data model is assumed to be either a linear or a nonlinear (with ReLU) model of inputs. The authors find that the student model can achieve a lower error than the weak teacher through early stopping, and they also identify the spectral conditions of the data covariance matrix, under which weak-to-strong generalization provably occurs.

**Strengths:**

1. The authors study an important and timely topic that is quite relevant to the ICLR community

2. The paper addresses a gap in the theoretical literature of weak-to-strong generalization: a theoretical characterization of the regime of underspecified teachers and overspecified students for weak-to-strong generalization.

3. The paper presents the topic and related work well. Additionally, the theoretical results are presented in an intuitive manner to facilitate understanding for the reader. Therefore, for the most part, the paper is well-written.

4. The authors provide experimental results beyond their synthetic setting in order to demonstrate the generality of their findings.

**Weaknesses:**

1.  The setting and theoretical analysis are too close to those in (Medvedev et al., 2025), while the fundamental distinction is that the current submission assumes underspecified teachers (cannot recover the ground-truth), which was not the case in (Medvedev et al., 2025). Although I acknowledge this distinction, I am not convinced about the novelty of this work, as some findings (e.g., weak-to-strong generalization is enabled by early stopping) also overlap with those of Medvedev et al. (2025). Even, most of the notations and analysis approaches used seem the same or very similar for these two papers. The authors should explicitly state the additional theoretical challenge posed by their setting and the distinct aspect of their findings in comparison to Medvedev et al. (2025).

2. The distinguishing finding of this work is unclear compared to the literature on weak-to-strong generalization.

3. The setting is limited since teachers and students are assumed to be random feature models, and the loss is MSE, limiting the impact of the findings. On the other hand, the literature moves towards theoretical characterizations of weak-to-strong generalization in feature learning regimes with nonlinear models (e.g., see the paper mentioned in the Minor below).

4. Related to weakness 3 (above): The studied teachers and students may be very close to linear models since there is no feature learning, and the input distribution is Gaussian with zero mean and diagonal covariance. If this is the case, it would further weaken the generality of the setting.

(*Explanation:* due to Gaussian universality (Hu and Lu, 2022), one can replace the random features $\sigma(x^T U)$ with a Gaussian vector with matched mean and covariance, while the test/generalization error is unchanged. This equivalent Gaussian model is of the form: $\mu_0 + \mu_1 x^T U + \mu_2 z$ for constants $\mu_0,\mu_1,\mu_2$ and $z \sim \mathcal{N}(0,I)$, which can be considered as a linear model with some Gaussian noise. For more, please see (Hu and Lu, 2022): Universality Laws for High-Dimensional Learning with Random Features.)

5. While the authors theoretically characterize the error (MSE), the theoretical results are not explicitly incorporated into the figures, weakening the connection between the theory and simulations.

6. The simulation results are given for a single random seed. The average of multiple Monte Carlo runs should be plotted for statistical significance.

7. Some experimental details are unclear. For example, see question 5 below.

8. Extension to real-world data is quite limited. In this regard, the authors provide only a table of three accuracy values for the MNIST dataset. These results just indicate that increasing teacher size leads to improved student performance, which is trivial.

**Questions:**

1. What are the additional theoretical challenges posed by the setting and the distinct aspects of the findings in comparison to the literature?

2. Do the findings extend to loss functions other than MSE?

3. (Regarding weakness 4) Do the considered teacher and student models outperform linear models?

4. Why haven't the authors incorporated the theoretical results into the figures?

5. In the setting of Figure 3 (Section 5.3), how are the weights for the ground-truth model selected?

6. Could the authors also demonstrate the findings in Figures 1-2 (which use synthetic data) using the MNIST dataset?

7. Could the authors explicitly state the spectral properties of the data covariance matrix that lead to weak-to-strong generalization?

**Minor:** There is a new paper that should be included in the related work: Oh et al., "From Linear to Nonlinear: Provable Weak-to-Strong Generalization through Feature Learning." NeurIPS 2025.

---

### Meta-Review · Area_Chair_4LPx · 2025-12-29

**Summary:**

This paper aims to provide a theoretical understanding of weak-to-strong generalization using random feature models. All reviewers consistently lean toward rejection, mainly due to an insufficient comparison with the closely related work of Medvedev et al. (2025). In particular, the setup seems very similar to Medvedev et al. (2025), with this paper placing additional emphasis on an underspecified teacher model. However, it remains unclear how the technical challenges, core insights, or main takeaways differ from those in Medvedev et al. (2025). Moreover, Reviewer DZyZ notes that some of the paper’s purportedly novel claims may be incorrect. Unfortunately, the authors did not submit a rebuttal to address these key concerns.

**Reviewer Concerns:**

No rebuttal was submitted, so the reviewers' concerns remain unaddressed.

**Reviewer Scores:**

Since the authors did not submit a rebuttal, it is very unlikely that any of the reviewers will change their score.

---

### Decision · Program_Chairs · 2026-01-26

Reject